# Somatodendritic orientation determines tDCS-induced neuromodulation of Purkinje cell activity in awake mice

**Carlos A Sánchez-León[1,2], Guillermo Sánchez-Garrido Campos[1], Marta Fernández[3,4], Álvaro Sánchez-López[5], Javier F Medina[5], Javier Márquez-Ruiz[1]***

[1]Department of Physiology, Anatomy and Cell Biology, Pablo de Olavide University, Ctra. de Utrera, Seville, Spain; [2]Department of Neurology and Neurobiology, University of California, Los Angeles, Los Angeles, United States; [3]Department of Psychiatry, University of California, Los Angeles, Los Angeles, United States; [4]Department of Pharmacology, University of the Basque Country (UPV/EHU), Leioa, Spain; [5]Department of Neuroscience, Baylor College of Medicine, Houston, United States

**\*For correspondence:**
jmarquez@upo.es

**Competing interest:** The authors declare that no competing interests exist.

## eLife Assessment

In this **important** and **compelling** study, Sánchez-León et al. investigate the effects of tDCS on the firing of single cerebellar neurons in awake and anesthetized mice. They find heterogeneous responses depending on the orientation of the recorded Purkinje cell. The paper may well explain part of the controversial and ambiguous outcomes of various clinical trials.

**Abstract** Transcranial direct-current stimulation (tDCS) of the cerebellum is a promising non-invasive neuromodulatory technique being proposed for the treatment of neurological and neuropsychiatric disorders. However, there is a lack of knowledge about how externally applied currents affect neuronal spiking activity in cerebellar circuits in vivo. We investigated how Cb-tDCS affects the firing rate of Purkinje cells (PC) and non-PC in the mouse cerebellar cortex to understand the underlying mechanisms behind the polarity-dependent modulation of neuronal activity induced by tDCS. Mice (n=9) were prepared for the chronic recording of local field potentials (LFPs) to assess the actual electric field gradient imposed by Cb-tDCS in our experimental design. Single-neuron extracellular recording of PCs in awake (n=24) and anesthetized (n=27) mice was combined with juxtacellular recordings and subsequent staining of PC with neurobiotin under anesthesia (n=8) to correlate their neuronal orientation with their response to Cb-tDCS. Finally, a high-density Neuropixels recording system was used to demonstrate the relevance of neuronal orientation during the application of Cb-tDCS in awake mice (n=6). In this study, we observe that Cb-tDCS induces a heterogeneous polarity-dependent modulation of the firing rate of PCs and non-PC in the mouse cerebellar cortex. We demonstrate that the apparently heterogeneous effects of tDCS on PC activity can be explained by taking into account the somatodendritic orientation relative to the electric field. Our findings highlight the need to consider neuronal orientation and morphology to improve tDCS computational models, enhance stimulation protocol reliability, and optimize effects in both basic and clinical applications.

## Introduction

Transcranial direct-current stimulation (tDCS) is a non-invasive brain stimulation technique consisting of the application of constant weak electric currents over the scalp for several minutes through strategically positioned electrodes (*Nitsche and Paulus, 2000*; *Woods et al., 2016*). In the last two

decades, tDCS has shown promising results in modulating cognitive, behavioral, and clinical traits by applying stimulation on various brain regions (*Stagg et al., 2018*). Cerebellar tDCS has been found to modulate both motor and also cognitive and emotional processing. This has led to its proposal as a noninvasive neuromodulatory therapy for treating cerebellum-related disorders (*Grimaldi et al., 2016*; *Manto et al., 2022*). As the popularity of tDCS grows, there are increasing debates regarding the variability in results among studies (*Antal et al., 2015*; *Horvath et al., 2015*; *Beyer et al., 2017*; *Jalali et al., 2017*). This unreliability can be attributed to methodological differences and the lack of knowledge about the physiological mechanisms of action of tDCS (*Horvath et al., 2016*; *Liu et al., 2018*; *Polanía et al., 2018*). To optimize the effects of tDCS in humans, it is essential to understand the neural basis underlying this variability and develop new strategies accordingly.

Animal models have been instrumental in advancing our understanding of the mechanisms underlying tDCS effects (*Jackson et al., 2016*), defining safety parameters (*Liebetanz et al., 2009*; *Jackson et al., 2017*), and inspiring new stimulation protocols (*Sánchez-León et al., 2018*). In vitro studies have highlighted the importance of neuronal orientation and morphology (*Bikson et al., 2004*; *Radman et al., 2009*; *Kabakov et al., 2012*) in modulating individual neurons' excitability. For neurons whose somatodendritic axis is aligned with the electric field, the field induces a pronounced somatic polarization. In the case of anodal stimulation, where the positive electrode is positioned near the dendrites and the soma is oriented away, positively charged ions accumulate near the soma, leading to depolarization and increased excitability, thus facilitating action potential generation. Conversely, neurons whose orientation opposes the field, such as when the soma is closer to the positive electrode and the dendrites face away, experience hyperpolarization, reducing excitability. Lastly, neurons oriented perpendicular to the electric field would exhibit minimal somatic polarization, as the field does not induce significant redistribution of charges along the somatodendritic axis. In addition, in vivo experiments also enable the study of the impact of tDCS on behaving brains and provide insights into the underlying neuronal mechanisms at both behavioral and physiological levels. For example, in vivo animal models have demonstrated the physiological effects of transcranial electrical stimulation (tES, encompassing tDCS and tACS) on spike timing (*Asan et al., 2020*; *Huang et al., 2021*), local field potential oscillations (*Krause et al., 2017*; *Sánchez-León et al., 2021*), memory and learning processes (*Márquez-Ruiz et al., 2012*; *Krause et al., 2017*), as well as the involvement of non-neuronal elements like astrocytes and microglial cells (*Monai et al., 2016*; *Mishima et al., 2019*). In vivo experiments also allow for the measurement of the actual intracranial electric field induced by tES (*Opitz et al., 2016*; *Sánchez-León et al., 2021*) and for testing the efficacy of tES protocols at both behavioral and physiological levels (*Grossman et al., 2017*; *Vöröslakos et al., 2018*). However, in vivo animal models often have limited control over the applied electric fields compared to in vitro experiments and may lack the ability to identify which specific cell types are impacted by the stimulation. Here, we overcome these obstacles by examining the impact of tDCS on the activity of identifiable neurons in the cerebellum of mice.

Purkinje cells (PCs) are the output neurons of the cerebellar cortex and can be electrophysiologically identified by their ability to fire both complex and simple spikes (SS) (*Thach, 1967*). Unlike the cerebral cortex, the cerebellar cortex of rodents is highly convoluted, similar to that of humans (*Ito, 1984*). In this study, the highly-folded mouse cerebellum was used to determine the impact of the PC orientation on the final neuronal modulation induced by tDCS in awake mice. We hypothesized that the orientation of PCs relative to the electric field would influence the effects of tDCS on neural activity. In the Vermis, PCs oriented parallel to the field are expected to exhibit stronger effects due to greater somatic polarization, leading to depolarization or hyperpolarization depending on the orientation of the somatodendritic axis. Conversely, PCs in Crus I/II, which are oriented obliquely to the field, are expected to exhibit intermediate effects, as the oblique alignment reduces the strength of polarization compared to parallel alignment. To test this hypothesis, single-neuron extracellular recording of PCs in awake mice was combined with juxtacellular recordings and subsequent staining of PC with neurobiotin under anesthesia to investigate the effects of cerebellar tDCS on the cerebellar network output. The morphological reconstruction of the recorded PCs allowed the correlation of their neuronal orientation with their response to tDCS. Finally, a high-density Neuropixels recording system was used to demonstrate the relevance of neuronal orientation in awake mice by simultaneously recording PCs with opposing orientations during the application of tDCS.

Our work offers in vivo evidence that points to neuronal orientation as a crucial factor in determining the impact of tDCS on neural activity, and can explain the reason why the effects are heterogeneous (or even opposite) across different layers of the same brain area. This result is essential for developing accurate computational models and emphasizes the need to consider neuronal orientation when predicting tDCS effects. Our findings suggest that high-definition tDCS electrodes in which the direction of the electric field can be flexibly controlled may be useful for enhancing the reliability of stimulation protocols and optimizing the desired tDCS effects in various cerebellum-related disorders (*Grimaldi et al., 2016*; *Manto et al., 2022*).

## Results
### Electric field measurement in the cerebellar cortex

tDCS effects critically depend on the strength of the electric field imposed in the brain. To assess the actual electric field gradient imposed by tDCS in the cerebellar cortex in our experimental design, a group of mice (n=9) was prepared for the chronic recording of LFPs in awake condition

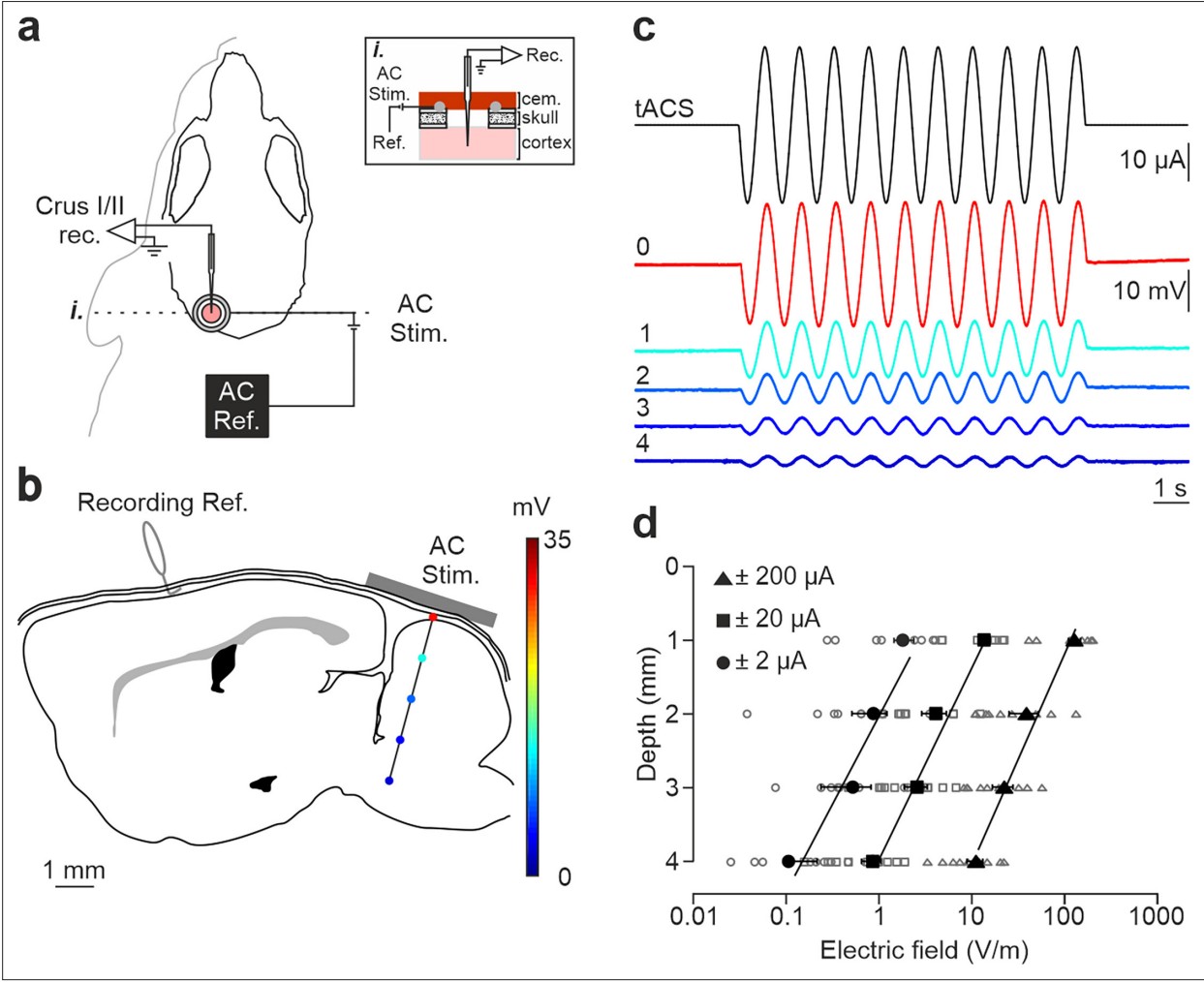

**Figure 1.** Intracranial electric fields induced by Cb-tACS. (**a**) Experimental design for combined in vivo electrophysiology and transcranial alternating current stimulation (tACS) in crus I-II of cerebellar cortex in awake mice, showing silver-ring active and reference (AC Ref.) electrode locations. Inset (i.) shows a schematic sagittal view of the recording chamber design. (**b**) Schematic representation of a sagittal section of the brain showing the reference used for electrophysiological recordings (Recording Ref.), the location of the active electrodes (gray bar), and a representative track in the lateral cerebellum highlighting the depths where the electric field was measured (color dots). (**c**) tACS stimulation (top trace) applied over the scalp and exemplary recording of the actual field potentials generated at different depths (from 0 to 4 mm) in a representative animal. The traces were overlapped to facilitate amplitude comparison. (**d**) Average (filled symbols) and individual (empty symbols) electric field strength recorded at different depths for ±2 (circles), ±20 (squares), and ±200 μA (triangles) tACS.

during simultaneous application of low-frequency transcranial alternating-current stimulation (tACS; 1 Hz) at different intensities (±2, ±20, and ±200 µA) (*Figure 1a*). LFPs were sequentially recorded every 1 mm from the cortical surface to 4 mm depth (15° rostro-caudal insertion angle; *Figure 1b*). *Figure 1c* shows the LFP recordings from a representative animal during simultaneous tACS (1 Hz, 10 s, ±20 µA) at different depths, showing a decrease in voltage for deeper recordings. The estimation of the electric field strength calculated at different depths (1, 2, 3, and 4 mm) and tACS intensities (±2, circles; ±20, squares; ±200 µA, triangles) is represented in *Figure 1d* (n=9). Under the active electrode, the magnitude of the electric field decreased with depth in a logarithmic manner for the three tested intensities (*R*=0.98, 0.96 and 0.95 for ±2, ±20, and ±200 µA, respectively; data is presented with logarithmic abscissa axis for visual facilitation, *Figure 1d*). With this data, the electric field imposed by tACS at different depths and intensities tested in our experiments was interpolated. A polynomial surface (degree 2 for depth and degree 2 for intensity axis) was fitted on the electric field values and then the coefficients (with 95% confidence bounds) were extracted from the linear model. Considering the most superficial (0.3 mm) and the deepest (2.3 mm) recorded neurons, we could expect electric field values between 60.1, 92.9 and 125.7 V/m (at 0.3 mm for 100, 200 and 300 µA, respectively) and 5.9, 20.2, and 34.6 V/m (at 2.3 mm for 100, 200, and 300 µA, respectively) at the recording places. These values are in line with in vitro and other in vivo animal studies showing modulation of neuronal firing rate under similar current densities (*Jackson et al., 2016*).

## tDCS modulates Purkinje cell activity in awake mice in a heterogeneous manner

To understand how tDCS modulates neuronal firing behavior at a single-cell level, we performed single-cell recordings in awake mice using glass micropipettes. We first focused on the potential impact of exogenous electric fields on PC firing rate. Unlike other cerebellar neuronal types, PCs can be electrophysiologically identified by the presence of high-frequency simple spikes (SS) and less frequent complex spikes (CS) followed by a brief SS silence (*Thach, 1967*; *Ito, 1984*; *Figure 2a and b*, *Figure 2—figure supplement 1a-d*). To determine the online effects of exogenous electric field application on PC firing rate and avoid long-term plasticity mechanisms, short pulses of tDCS (15 or 20 s including 5 s ramp-in and 5 s ramp-out, ±200 µA) separated by non-stimulating periods (10 s) were applied. This protocol allows us to avoid the development of plasticity effects, which are known to require at least several minutes of tDCS administration, and to test the direct electrical modulation exerted by the externally applied currents. *Figure 2a–f* shows two representative PCs and their firing behavior during tDCS (*Figure 2c–f*). As observed in the Z-score-transformed average firing rate, the SS firing rate of some recorded PCs significantly increased during anodal and decreased during cathodal tDCS (*Figure 2c and e*), whereas the opposite effects were observed in other recorded PCs; decreasing the SS firing rate during anodal and increasing during cathodal (RM-ANOVA or Friedman tests, p<0.01) (*Figure 2d and f*). Furthermore, we could observe a significant rebound effect when tDCS was switched off for some of the neurons (*Figure 2d and f*, 'After'). A total of 24 identified PCs were recorded in crus I-II region (*Figure 2g and h*) in awake mice (n=24 animals). The impact of anodal and cathodal tDCS on the SS firing rate showed significant differences (filled circles) for 11 out of 24 individual PCs recorded (n=24, RM-ANOVA or Friedman tests, p<0.05) (*Figure 2g*, *Figure 2—figure supplement 2a, b*). The data distribution was fitted by a linear model (*R*=0.46, p=0.0003), which indicates that most of PCs increased their firing rate with anodal stimulation and decreased with cathodal stimulation (cells in 3rd quadrant of *Figure 2g*) or vice versa (cells in 1st quadrant of *Figure 2g*).

No differences were observed in the waveform of recorded SS nor CS during anodal nor cathodal tDCS with respect to control condition (*Figure 2—figure supplement 1a–d*). No significant changes were observed in the CS firing rate during tDCS (*Figure 2—figure supplement 1e–h*; n=25, RM-ANOVA or Friedman tests) except for one of the recorded PCs. The impact of anodal and cathodal tDCS in CS firing rate and the CS-driven SS silence of the individual PCs recorded in crus I-II of awake mice was not significant for most cells (*Figure 2—figure supplement 1i and j*).

From these results, we can conclude that PCs, constituting the only output from cerebellar cortex, (1) are modulated by tDCS based on the electric field gradient applied in this experiment and (2) this modulatory effect is heterogeneous across the PC population.

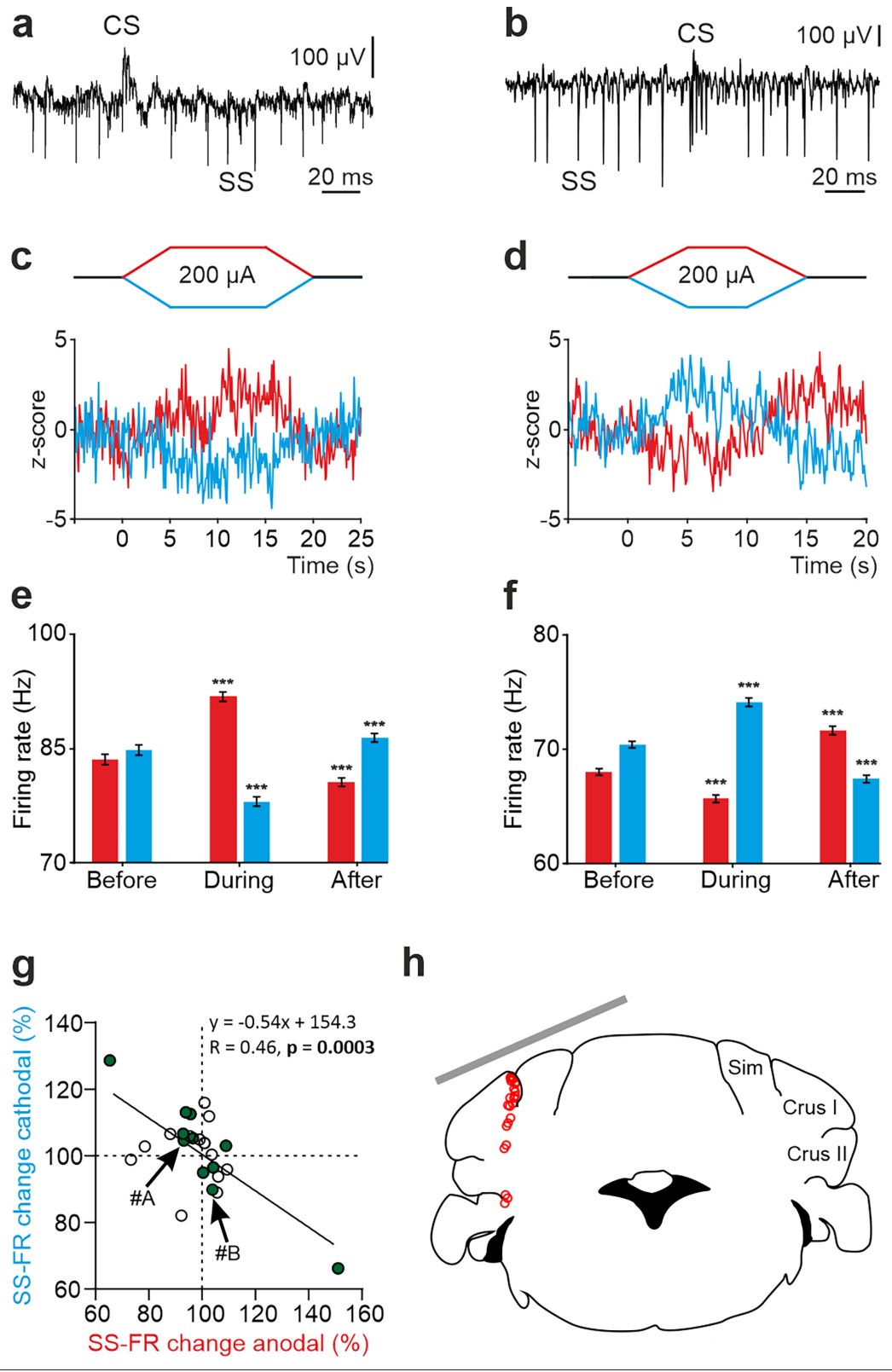

**Figure 2.** tDCS modulation of PC activity in crus I/II of awake mice. (**a**, **b**) Recording of spontaneous firing activity of two PCs showing the presence of SS and CS. (**c,d**) Z-score-transformed average PSTH (bin size: 0.1 s) of the SS activity of the 2 PCs shown in a,b before, during and after anodal (red trace) or cathodal (blue trace) tDCS. (**e,f**) Statistical comparison of the SS firing rate of the 2 PCs shown in a,b, measured in 5 s windows before,

*Figure 2 continued on next page*

*Figure 2 continued*

during and after tDCS (**e**): Anodal: RM-ANOVA, F (3, 196)=61.305, N=50, p<0.001; Cathodal: Friedman, $\chi^2$ (3, 147)=76.316, N=50, p<0.001; (**f**): Anodal: Friedman, $\chi^2$ (2, 98)=68.593, N=50, p<0.001; Cathodal: RM-ANOVA, F (2, 147)=107.859, N=50, p<0.001. Error bars represent SEM. **p<0.01; ***p<0.001. (**g**) Modulation of SS of individual PCs (circles) during anodal (red) and cathodal (blue) tDCS. Filled circles represent statistically significant modulation during tDCS (n=24, RM-ANOVA or Friedman tests, p<0.05). Arrows indicate data from the example neurons shown in panels A and B. (**h**) Schematic representation of the recording sites and active electrode location (gray bar) during tDCS.

The online version of this article includes the following figure supplement(s) for figure 2:

**Figure supplement 1.** tDCS does not modulate PC waveform or complex spikes in the awake mouse.

**Figure supplement 2.** Summary of firing rate modulation for all recorded PCs and non-PCs during tDCS over crus I-II cerebellar region.

## Non-Purkinje cell activity is also modulated by tDCS in awake mice

Beyond the inhibitory PCs, the cerebellar cortex neuronal network also contains excitatory (granule cell) and inhibitory (Golgi cell, Lugaro cells, basket cells, and stellate cells) neurons responsible for determining the spatio-temporal PC output (*Prestori et al., 2019*). The modulatory effects of tDCS on the activity of these non-PC neurons could be important in feed-forward, feedback, and lateral inhibition processes underlying cerebellar function. Unlike PCs, these neurons do not show typical SS and CS in the neuronal recording and cannot be electrophysiologically identified. For analysis purposes, we decided to include these neurons in 'non-PC' group. The same tDCS protocol previously used with PCs was applied to determine the online effects of exogenous electric field application on non-PC firing rate.

*Figure 3a–f* shows two representative non-PCs and their firing behavior during tDCS. As observed in the Z-score-transformed average, the firing rate of some recorded non-PCs significantly increased during anodal and decreased during cathodal tDCS (*Figure 3c,d*) whereas different effects were observed in other recorded neurons (RM-ANOVA or Friedman tests, p<0.05) (*Figure 3e and f*). Furthermore, similar to what we previously observed in PCs, we could observe a significant rebound effect when tDCS was switched off for some of the neurons (*Figure 3c-f*). A total of 50 non-PCs were recorded in crus I-II region (*Figure 3g and h*) in awake mice (n=24 animals). The impact of anodal and cathodal tDCS in the firing rate showed significant differences (filled circles) for 35 out of 50 individual non-PCs recorded (n=50, RM-ANOVA or Friedman tests, p<0.05) (*Figure 3g*, *Figure 2—figure supplement 2c and d*). The data distribution was fitted by a linear model (*R*=0.56, p<0.0001) and was found to have higher dispersion than the PC data (*Figure 2g*). No differences were observed in the waveform of recorded spikes during anodal nor cathodal tDCS with respect to control condition (data not shown).

These results allow us to conclude that not only PCs but also non-PCs implicated in the spatio-temporal response of PCs are modulated during tDCS in a heterogeneous way.

## Purkinje cell orientation explains heterogeneous tDCS modulation in anesthetized mice

Given the large heterogeneity observed in the responses of the recorded cerebellar neurons to tDCS and considering the anatomical complexity of the highly convoluted cerebellar cortex, we wondered if the somatodendritic axis orientation of the cerebellar neurons could partially explain this variability, as suggested by previous in vitro studies for neurons in other brain regions (*Bikson et al., 2012*; *Kabakov et al., 2012*; *Radman et al., 2009*). For this purpose, we decided (1) to record in the cerebellar vermis region which is characterized by having PCs oppositely oriented in adjacent cortical layers (*Figure 4a and b*) and (2) label some of the recorded PC with neurobiotin after electrophysiological characterization to examine their orientation.

A total of 56 neurons (31 identified PCs and 25 non-PCs) were recorded in the vermis of 27 anesthetized mice (*Figure 4a and b*) during anodal and cathodal tDCS (±200 µA). The impact of anodal and cathodal tDCS on the firing rate showed significant differences (filled circles) in 27 out of 31 PCs (*Figure 4c*) and in 17 out 25 recorded non-PCs (*Figure 4d*) (n=31 PCs and 25 non-PCs, RM-ANOVA or Friedman tests, p<0.05). The effects of tDCS on PC firing rate adjusts to a linear regression (slope

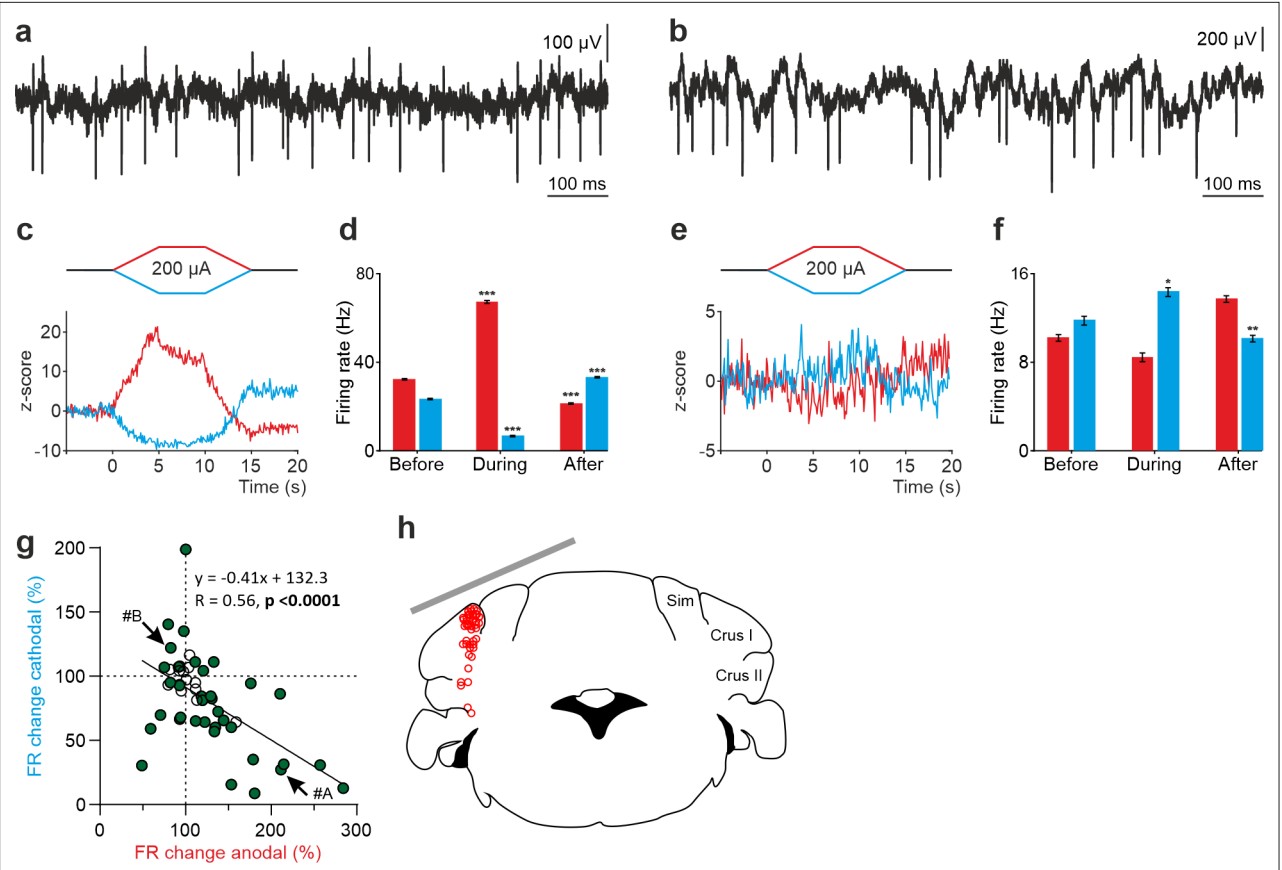

**Figure 3.** tDCS modulation of non-PC activity in crus I/II of awake mice. (**a**, **b**) Recording of spontaneous firing activity of two non-PCs. (**c, e**) Z-score-transformed average PSTH (bin size: 0.1 s) of firing rate for the 2 neurons in a,b, before, during and after anodal (red trace) or cathodal (blue trace) tDCS. (**d, f**) Statistical comparison of the firing rate of the 2 neurons in a,b, measured in 5 s windows before, during and after tDCS (**d**): Anodal: Friedman, $\chi^2$ (2, 98)=100, N=50, p<0.001; Cathodal: Friedman, $\chi^2$ (2, 98)=100, N=50, p<0.001; (**f**): Anodal: RM-ANOVA, F (2, 147)=64.656, N=50, p<0.001; Cathodal: Friedman, $\chi^2$ (2, 98)=36.765, N=50, p<0.001. Error bars represent SEM. *p<0.05; **p<0.01; ***p<0.001. (**g**) Modulation of firing rate of individual neurons (circles) during anodal (red) and cathodal (blue) tDCS. Filled circles represent statistically significant modulation during tDCS (n=50, RM-ANOVA or Friedman tests, p<0.05). Arrows indicate data from the example neurons shown in panels A and B. (**h**) Schematic representation of the recording sites and active electrode (gray bar) location during tDCS.

–0.57, R=0.7, p<0.0001) (*Figure 4c*), whereas the effects were scattered on non-PCs (slope –0.19, R=0.35, p<0.0875) (*Figure 4d*).

This result suggests that in the vermis there is a consistent polarity-dependent modulation for PCs, where anodal and cathodal modulate firing rate in opposite ways. However, this effect is not observed in the non-PC group. The findings remained consistent even when we tested the same neurons with greater (±300 µA) and lesser (±100 µA) current intensities (*Figure 4—figure supplement 1*). Finally, we compared the modulation between all PC recorded in Vermis or Crus I/II. To this end, we normalized the modulation so that all data are positive. For example, a neuron that increases or decreases its activity by 50% relative to the baseline period will be considered as having a modulation of 50% in both cases. This yields a mean modulation of 62.35% for neurons recorded in Vermis and 9.42% for those in the Crus I/II. Statistical comparison between the neurons modulated in both regions (Mann-Whitney test) resulted in a p-value <0.001, corroborating that the modulation exerted by tDCS in the Vermis is greater than that exerted when applied over Crus I/II. From these experiments, we can conclude that (1) tDCS in vermis of anesthetized mice modulates PCs and non-PCs in a heterogeneous way, (2) tDCS in vermis modulates more PCs than in Crus I-II, (3) most PCs in vermis modulate in opposite directions for anodal vs cathodal tDCS and (4) tDCS over vermis modulates PCs more than over Crus I/II.

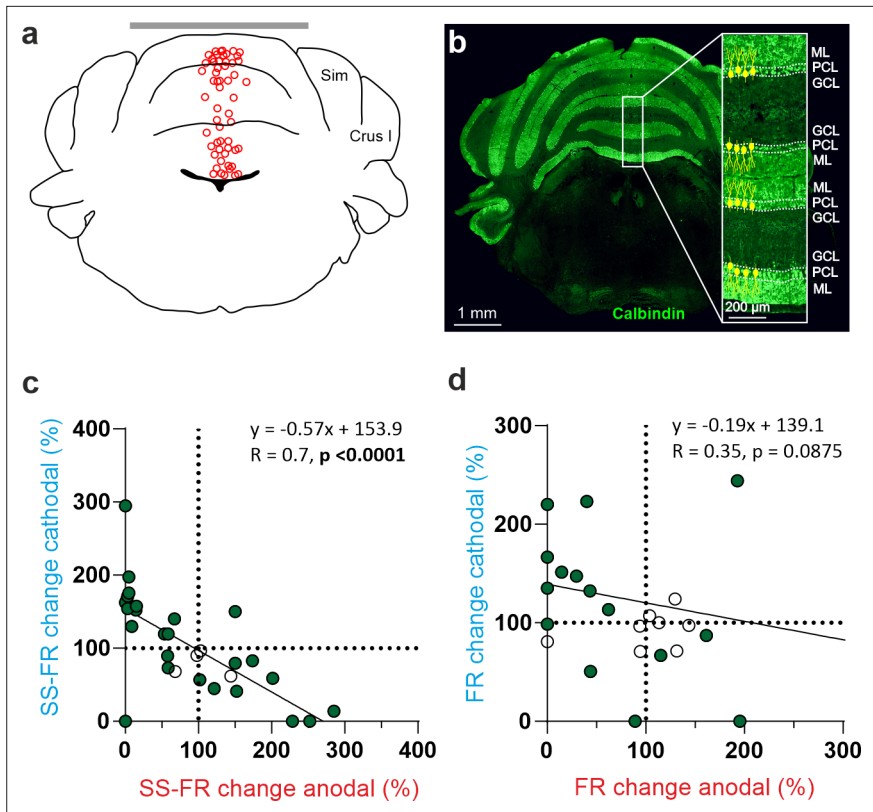

**Figure 4.** tDCS modulation of PC and non-PC activity in the vermis of anesthetized mice. (**a**) Schematic representation of the recording sites and active electrode (gray bar) location during tDCS. (**b**) A representative coronal section of the vermis immunofluorescently stained with Calbindin (green). The magnification inset highlights the distinct orientation of PCs in different layers, which is indicated by the drawings of PC somas and dendrites (with dendrites always extending into the molecular layer, shown in green). (**c,d**) Modulation of SS firing rate of individual PCs (**c**) and firing rate of individual non-PCs (**d**) during anodal (red) and cathodal (blue) tDCS over cerebellar vermis. Filled circles represent statistically significant modulation during tDCS (n=31 PCs and 25 non-PCs, RM-ANOVA or Friedman tests, p<0.05). GCL: granular cell layer, ML: molecular layer, PCL: Purkinje cell layer.

The online version of this article includes the following figure supplement(s) for figure 4:

**Figure supplement 1.** tDCS modulation of PC and non-PC activity at different intensities in anesthetized mice.

---

To test if the opposite modulation of PCs by anodal and cathodal stimulation could be related to the opposite somatodendritic orientation of these neurons with respect to the electric field, a total of eight recorded PCs in eight animals were successfully stained with neurobiotin using juxta-cellular microinjection in the anesthetized mice either in crus I-II or vermis (see Methods). Labeled PCs were reconstructed with confocal microscopy and the deviation of the somatodendritic axis from the imaginary line perpendicular to the skull surface under the active electrode was calculated (*Figure 5a*; $\theta$ angle). *Figure 5a–d* shows representative PCs with different $\theta$ angle together with Z-score-transformed average PSTH (bin size: 0.1 s) of the spiking activity and statistical comparison of the firing rate before, during and after anodal (red) or cathodal (blue) tDCS (Friedman test, p<0.05). We observed that when $\theta$ was close to 0° (the somatodendritic axis was pointing toward the active electrode, *Figure 5a*) anodal tDCS tended to increase the firing rate and cathodal reliably decreased it, whereas when $\theta$ was close to 180° (the somatodendritic axis was pointing away from the electrode; *Figure 5b*), the opposite modulation was observed, with anodal decreasing and cathodal increasing the firing rate. However, when $\theta$ value was close to 90° (*Figure 5c*) or to 270° (*Figure 5d*; the axis was more perpendicular to the electric field) the modulatory effect was absent. *Figure 5e* summarizes the relationship between the $\theta$ angle and tDCS modulation of all labeled neurons. The figure represents the normalized firing rate modulation ([firing rate during tDCS/firing rate before tDCS]*100; length of the arrow) and $\theta$ angle value for each recorded neuron during anodal (*Figure 5e*, at left) and cathodal

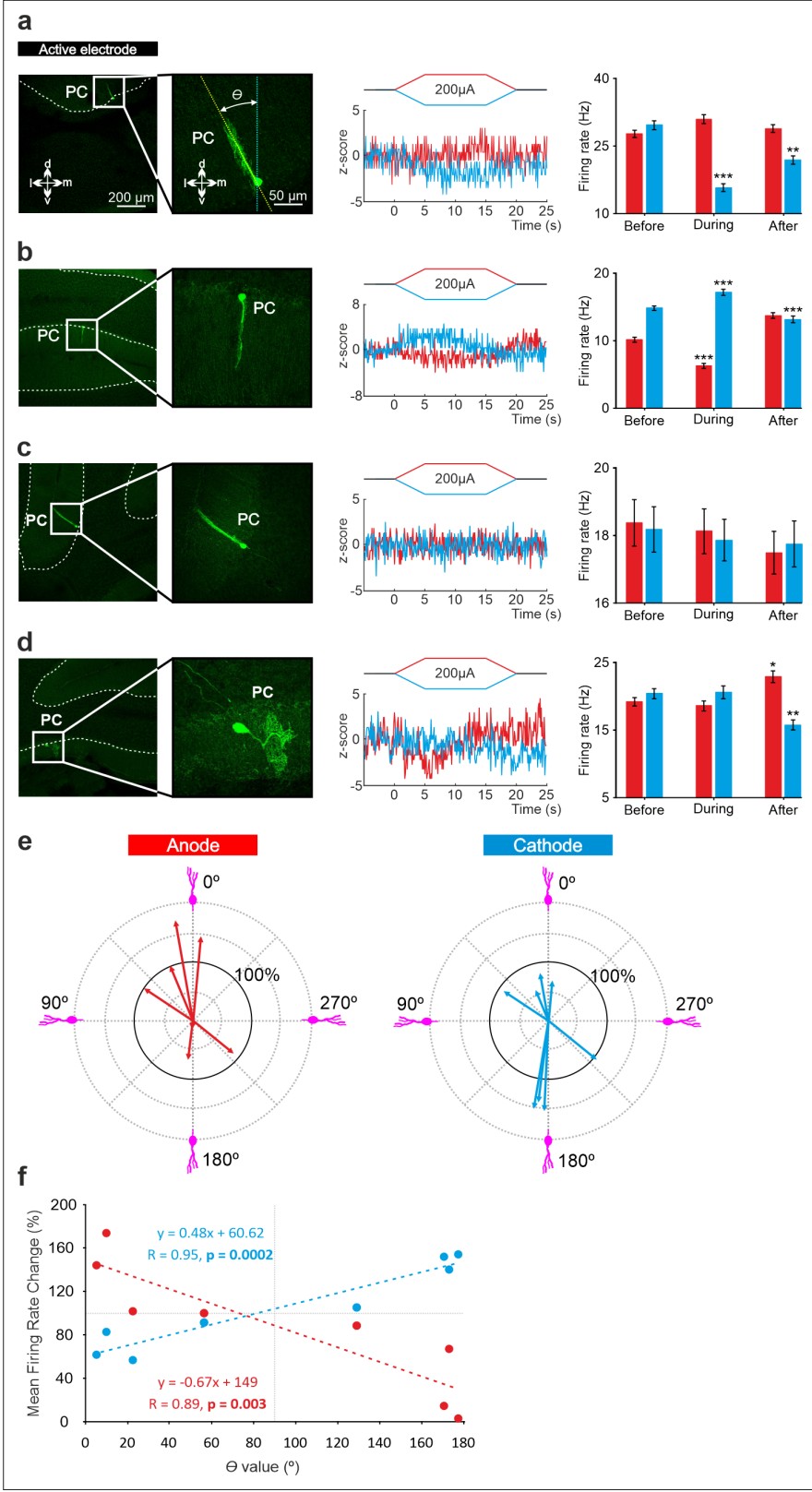

**Figure 5.** Relationship between tDCS-driven modulation of PC firing rate and somatodendritic axis orientation in anesthetized mice. (**a**-**d**) (Left) Confocal images of labeled neurons with different somatodendritic angles relative to the electric field (dotted white vertical line), (Right) z-score of their firing rate modulation during anodal (red) or cathodal (blue) tDCS and statistical analysis (**a**): Anodal: Friedman, $\chi^2$ (3, 147)=5.482, N=50, p=0.139;

*Figure 5 continued on next page*

*Figure 5 continued*

Cathodal: Friedman, $\chi^2$ (3, 147)=71.544, N=50, p<0.001; (**b**): Anodal: Friedman, $\chi^2$ (3, 147)=93.289, N=50, p<0.001; Cathodal: Friedman, $\chi^2$ (3, 147)=81.428, N=50, p<0.001; (**c**): Anodal: Friedman, $\chi^2$ (3, 147)=2.348, N=50, p=0.503; Cathodal: Friedman, $\chi^2$ (3, 147)=3.065, N=50, p=0.382; (**d**): Anodal: Friedman, $\chi^2$ (3, 147)=29.904, N=50, p<0.001; Cathodal: Friedman, $\chi^2$ (3, 147)=23.385, N=50, p<0.001. Error bars represent SEM. *p<0.05; **p<0.01; ***p<0.001. (**e**) Relationship between firing rate modulation and somatodendritic angle for all juxtacellularly-labeled PCs (n=8). Arrow length represents firing rate modulation during anodal (red arrows, at left) or cathodal (blue arrows, at right) tDCS at 200 µA, relative to the firing rate during control condition (represented by 100% solid circle). Densely packed lines and suppressed activity of two neurons under anodal tDCS reduce the visibility of their responses. (**f**) Average change in firing rate during anodal (red) and cathodal (blue) tDCS for individual PCs with different somatodendritic orientations.

(*Figure 5e*, at right) tDCS. *Figure 5f* represents the relationship between the normalized firing rate modulation of each individual neuron with respect to its corresponding $\theta$ angle for anodal (red line) or cathodal (blue line) tDCS. The impact of tDCS in the firing rate of PC was higher for those neurons with $\theta$ values close to 0° and 180° for anodal and cathodal currents acting in an opposite polarity-dependent manner. These results corroborate that the somatodendritic axis orientation plays a critical role in explaining tDCS heterogeneous modulation of individual PCs in anesthetized mice.

## Purkinje cell orientation determines polarity of tDCS-driving firing rate modulation in awake mice

To examine whether the effects of PC orientation that we observed in anesthetized mice can be extended to the more clinically relevant awake state, we took advantage of Neuropixels technology and prepared six additional mice for chronic multiunitary neuronal recording during simultaneous tDCS. Neuropixels probes were coated with lipophilic dye (DiI) for subsequent histological reconstruction of the recording track (see Methods). A total of nine PCs were selected for analysis because they were particularly stable and well isolated during the entire tDCS session, and because they were unequivocally identified as PCs by confirming that they fired SS and CS. *Figure 6a* illustrates probe location marked with DiI in the cerebellar vermis of a representative animal where two different PCs with the somatodendritic axis pointing toward and away from the active electrode recorded at Ch#55 and Ch#42, respectively (*Figure 6b*); note dendrites always extend into the molecular layer which is shown in green. The Z-score-transformed average PSTH (bin size: 0.1 s for SS) of the SS activity before, during and after anodal (red trace) and cathodal (blue trace) tDCS is shown for these two simultaneously recorded PCs in *Figure 6c*. Remarkably, we found that the two PCs, which have oppositely oriented dendrites, exhibit opposite effects in presence of anodal or cathodal tDCS (*Figure 6c*). The same relationship was found for all the other PCs (*Figure 6*, *Figure 6—figure supplement 1*), where PCs in which the dendrites pointed toward the active tDCS electrode had increased firing rate during anodal tDCS and reduced firing rate during cathodal tDCS (*Figure 6d and e*, red triangles pointing up) while PCs with dendrites pointing away from the active tDCS electrode had the opposite modulation (*Figure 6d and e*, blue triangles pointing down). This experiment corroborates in the awake animal and in simultaneously recorded PCs that (1) a given tDCS polarity (either anodal or cathodal) can modulate individual PC neurons in opposite ways at the same time, increasing and decreasing their firing rates, and (2) the somatodendritic axis orientation of PCs is a key factor in determining the tDCS-driven modulation of firing rate.

## Discussion

In the present investigation, we use in vivo unitary extracellular recordings to show tDCS capability to modulate neuronal activity in the cerebellar cortex of awake mice. We find that both anodal and cathodal tDCS modulates the activity of many cerebellar neurons, but the effects are extremely heterogeneous. For PCs, which are the neurons responsible for sending the output of the entire cerebellar cortex, we show that the diverse effects observed during tDCS are largely explained by the somatodendritic axis orientation with respect to the active electrode. This is shown by matching in vivo electrophysiological recordings with neurobiotin-labeling of individual PC neurons in anesthetized mice, as well as Neuropixels high-density recordings in awake mice under tDCS. The firing rate of PCs

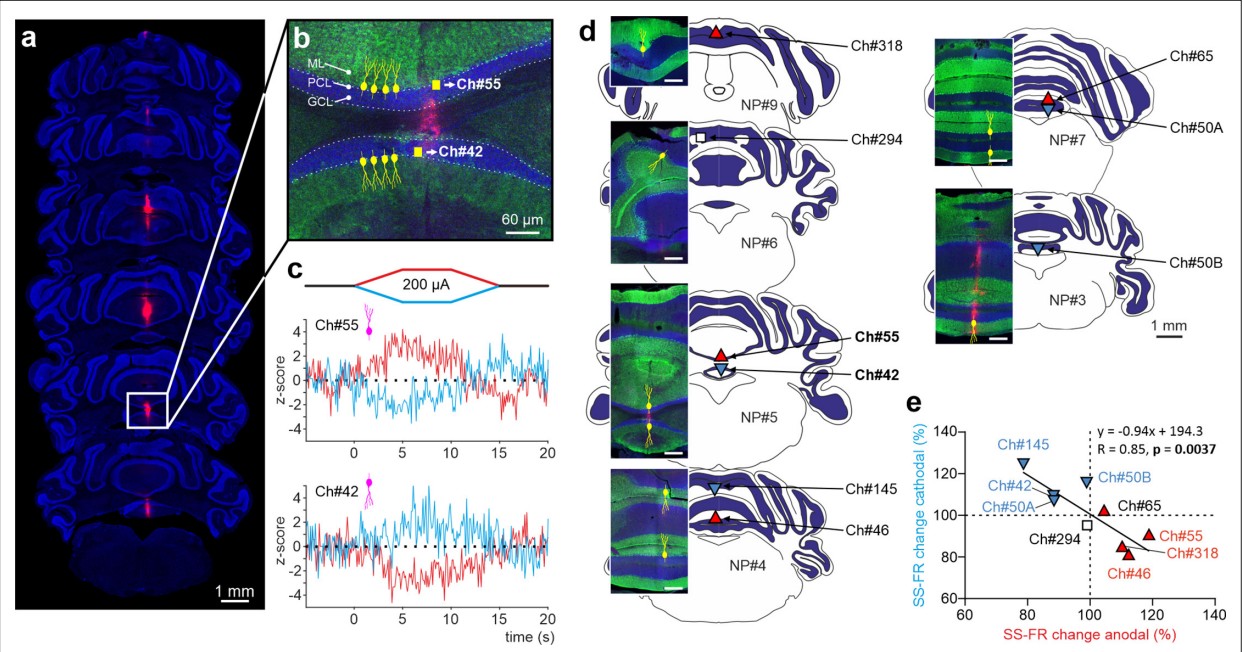

**Figure 6.** Impact of tDCS on PCs with opposite somatodendritic orientations simultaneously recorded in the awake mice. (**a**) Probe location marked with Dil in the cerebellar vermis stained with Hoechst 33342 dye. (**b**) Magnification of square area in '*a*' showing the location of two oppositely oriented PCs recorded at Ch#55 and Ch#42. The orientation of the PCs in each of the two layers is indicated with drawings of PC soma and dendrites (yellow), in which the soma appears at the interface between the granule cell layer (GCL, shown in blue) and the molecular layer (ML, shown in green). (**c**) Z-score-transformed average PSTH (bin size: 0.1 s for SS) of the SS activity before, during and after anodal and cathodal tDCS pulses for each of the two simultaneously recorded PCs shown in b (Ch#55: Anodal: Friedman, $\chi^2$ (2, 98)=67.647, N=50, p<0.001; Cathodal: Friedman, $\chi^2$ (2, 98)=54.576, N=50, p<0.001; Ch#42: Anodal: RM-ANOVA, F (2, 147)=123.947, N=50, p<0.001; Cathodal: RM-ANOVA, F (2, 147)=63.076, N=50, p<0.001). (**d**) Anatomical localization of the different PCs recorded. The inset shows the recorded places marked with Dil (red) and stained with anti-Calbindin antibody (green, molecular layer) and Hoechst 33342 (blue, granule layer; scale bar: 300 µm). PCs in which dendrites are pointed toward or away from the active tDCS electrode are denoted with triangles pointing upward or downward respectively. The color of the triangle indicates whether the modulation was an increase (red) or a decrease (blue) in firing rate. (**e**) Modulation of SS firing rate of individual PCs during anodal and cathodal tDCS. Filled symbols represent statistically significant modulation during tDCS, with the meaning of color and shape as in d (n=9, RM-ANOVA or Friedman tests, p<0.05). ML: molecular layer, PCL: Purkinje cell layer, GCL: granular cell layer.

The online version of this article includes the following figure supplement(s) for figure 6:

**Figure supplement 1.** tDCS modulation of PCs at different PC layers in the awake mice.

whose dendrites are pointed toward the stimulation electrode increases (or decreases) during anodal (or cathodal) tDCS, respectively, while the modulation of firing rate is exactly the opposite for PCs whose dendrites are pointed away from the stimulation electrode.

We observe a robust firing rate modulation during tDCS, with over 64% of the recorded neurons being modulated. The neuromodulatory effects of tDCS are highly dependent on the magnitude of the electric fields applied to the skull and the distance to the site of action in the brain (***Opitz et al., 2015***). As expected from cerebellar modelling studies (***Parazzini et al., 2014***; ***Rahman et al., 2014***) and previously observed in vitro (***Chan and Nicholson, 1986***) and in vivo (***Asan et al., 2020***), we find that the current flow in the cerebellum is largely uniform in direction, with the highest electric field values observed in the first millimeter of the cortex (64.8, 6.9 and 0.97 V/m for 200, 20 and 2 µA, respectively), and that the electric field strength decays logarithmically with distance. This electric field magnitude is directly related to the intensity of the electric current and the size of the electrodes, which determine the current density, as well as the shunting effect caused by the tissue between the electrodes and the brain (***Opitz et al., 2017***; ***Gomez-Tames et al., 2019***). Approximately 60% of the applied current is shunted by the skin, while around 20% is attenuated by the skull (***Liu et al., 2018***; ***Vöröslakos et al., 2018***). Based on previous studies that rely on computational analysis with realistic head models to estimate how electric currents propagate across the brain in different tDCS montages (***Parazzini et al., 2014***; ***Ruffini et al., 2014***; ***Huang et al., 2019***), we estimate that the electric fields

applied in this study are generally higher than those typically used in human protocols (~1–1.5 V/m), although they are similar to those used in other animal studies (*Jackson et al., 2016*; *Alekseichuk et al., 2019*; *Asan et al., 2019*). We used higher values than those applied in human experiments to achieve more reliable results. As seen in *Figure 4—figure supplement 1*, neurons are modulated in a similar way for 100, 200, or 300 µA but higher intensities elicited significant changes in a greater proportion of these neurons. In addition, a previous study from our lab (*Sánchez-León et al., 2021*) using the same methodology, showed that 200 µA (eliciting from 20.2 to 92.9 V/m in the current study) were ideal to obtain reliable and robust results in neuronal modulation, while keeping animal awareness of the stimulation at a minimum level. Besides, *Asan et al., 2019* has recently shown that using epidural stimulation in anesthetized rats under an electric field closer to human studies (1.5–7.5 V/m) was also able to modulate the activity of cerebellar neurons. Importantly, we observe clear firing rate modulation of PCs and non-PCs at depths of 2.3 mm and tDCS intensity of 100 µA, where the measured electric field is as low as 5.9 V/m.

The tDCS-driven modulation of firing rate is polarity-dependent for the majority of cerebellar neurons, with anodal and cathodal tDCS modulating the firing rate in opposite directions. Similar polarity-dependent modulation has been reported in previous in vivo studies in anaesthetized animals during application of tDCS over cerebral (*Bindman et al., 1964*; *Vöröslakos et al., 2018*) and cerebellar (*Asan et al., 2020*) cortices. The mechanism behind the firing rate changes observed during tDCS is likely due to a polarizing effect, where a slight alteration in the resting membrane potential of neurons (depolarization or hyperpolarization, depending on tDCS polarity) will lead to opposite changes in postsynaptic somatic spiking, as demonstrated in vitro (*Bikson et al., 2004*; *Farahani et al., 2021*) and in vivo (*Vöröslakos et al., 2018*). Consistent with this explanation, we observed that the firing rate is modulated over time in accordance with transcranial current dynamics (i.e. variable in ramp-in and ramp-out periods and stable during the 5 or 10 s at maximum intensity), except for a rebound effect observed after current termination (also observed by *Asan et al., 2020*). Nonetheless, we cannot rule out the possibility of indirect synaptic effects. Indeed, the electric field gradient imposed by tDCS could indirectly modulate a specific neuron firing rate by increasing (or decreasing) its pre-synaptic activity, that is by modulating the firing rate of other neurons that synapse onto it. Indeed, these synaptic changes could explain the rebound effect observed after tDCS termination. The synapses involved in the modulation of firing rate may undergo a short-term plasticity process (*Cirillo et al., 2017*; *Kronberg et al., 2017*; *Kronberg et al., 2020*; *Farahani et al., 2021*), which can continue to affect the firing rate even after the external currents have been turned off and no polarization is exerted on the neuron. These findings are consistent with previous work suggesting that synaptic plasticity is crucial for the effects of tDCS, as demonstrated by the importance of PC plasticity in behavioral outcomes (*Das et al., 2017*) and the role of BDNF-mediated plasticity in motor learning (*van der Vliet et al., 2018*).

Previous in vitro, ex vivo and modelling studies have highlighted the importance of various neuronal features that underly tDCS effects, including the orientation of somatodendritic axis with respect to the electrical field (*Rahman et al., 2014*), the neuronal morphology (*Chan and Nicholson, 1986*; *Radman et al., 2009*) or the axonal orientation (*Kabakov et al., 2012*). In this study, we demonstrate that in both anesthetized and awake animals, the orientation of the PC somatodendritic axis with respect to the electric field induced by tDCS explains the observed polarity-dependent heterogeneity in firing rate modulation. The importance of PC somatodendritic orientation in determining the effect of tDCS on firing rate modulation is further highlighted by our observation that pre-synaptic non-PC neurons providing inputs to PCs modulate their activity in a very heterogeneous way. In other words, our findings reveal that regardless of how tDCS impacts the activity of pre-synaptic inputs to PCs, the tDCS-driven firing rate modulation of each individual PC can be predicted by simply taking into account the orientation of the PC's dendrites relative to the electric field.

Our findings may provide some biological insight as to why the effects of cerebellar tDCS on motor control and learning have been difficult to replicate and are often reported to be fickle and unreliable (*Beyer et al., 2017*; *Jalali et al., 2017*). First, we find that in most of the analyzed PCs, tDCS induces a highly variable and inconsistent modulation of two parameters that play a key role in cerebellar learning function (*Medina and Lisberger, 2008*): the firing rate of PC CS and the duration of the SS silence following the CS. Second, we show that in the same animal and at the same time, tDCS can drive completely opposite changes in the firing rate of PCs with opposite somatodendritic

orientations. Groups of PCs in different layers and areas of the cerebellar cortex are linked together into functional modules, based on their projecting to the same subdivision of the deep cerebellar nuclei (*Apps and Hawkes, 2009*). As a result, the final macroscopic effect of cerebellar tDCS will depend on the net modulation of all the PCs in each module within the stimulated folia, which will be strongly influenced by their (likely heterogeneous) orientation. In the future, new technological advances in high-definition tDCS may be combined with flexible control of electric field direction to target PCs in specific modules and boost reliability.

In conclusion, we found that tDCS modulates the firing rate of PCs in mice in a polarity-dependent manner, and this modulation is highly dependent on the orientation of the PC somatodendritic axis relative to the electric field. Our findings emphasize the importance of considering the neuronal orientation and morphology of target neurons when applying transcranial stimulation, at least in the cerebellum. Taking into account, these neuronal features is crucial for increasing the predictive power of computational models and optimizing the desired effects of tDCS in basic and clinical human applications.

## Methods
### Animal preparation
Experiments were carried out on adult C57 male mice (n=74; University of Seville, Spain) weighting 28–35 g. All experimental procedures were carried out in accordance with European Union guidelines (2010/63/CE) and Spanish regulations (RD 53/2013) for the use of laboratory animals in chronic experiments. In addition, these experiments were submitted to and approved by the local Ethics Committee of the Pablo de Olavide University (Seville, Spain). Mice were prepared for simultaneous tES administration and electrophysiological recordings in the lateral (left) or vermis region of the cerebellar cortex in the head-restrained awake animal, following surgical procedures described previously (*Sánchez-León et al., 2021*). In brief, animals were anesthetized with a ketamine–xylazine mixture (Ketaset, 100 mg/ml, Zoetis, NJ., USA; Rompun, 20 mg/ml, Bayer, Leverkusen, Germany), and a custom-made chlorinated silver ring electrode (2.5 mm inner ø, 3.5 mm outer ø) was placed over the skull centered on left crus I-II (AP = − 6 mm; L = +2 mm; relative to bregma *Paxinos and Franklin, 2013*; *Figure 1a*) or on the cerebellar vermis (AP = − 6 mm; L=0 mm; relative to bregma) and fixed with dental cement (DuraLay, Ill., USA). A 2 mm ø craniotomy was made centered in the ring and exposing the cerebellar cortex. The dura was left intact and protected with wax bone (Ethicon, Johnson & Johnson) until recordings begun. In addition, a silver wire electrode (ø: 381 µm, A-M Systems) was also implanted over the dura surface under the left parietal bone (AP = − 0.9 mm; L = +3 mm; relative to bregma) as electrical reference for the electrophysiological recordings. Finally, a head-holding system was implanted, consisting of three bolts screwed to the skull and a bolt placed over the skull upside down and perpendicular to the frontal plane to allow for head fixation during the experiments. The holding system was cemented to the skull.

### Single unit recordings
Recording sessions began at least 2 days after surgery. The animals were placed over a treadmill with an infrared sensor for locomotion activity monitoring and the head was fixed to the recording table by means of the implanted head-holding system. Bone wax was removed with the aid of a surgical microscope (SMZ-140, Motic, Barcelona, Spain) and the cortical surface was carefully cleaned with super fine forceps (Dumont #5, FST, Heidelberg, Germany) and cotton swab without damaging the dura mater.

All single-cell recordings were carried out with an amplifier (BVC-700A, Dagan corporation, MN., USA) connected to a dual extracellular-intracellular headstage (8024, Dagan corporation, MN., USA; gain error ±1%, noise 4.5 µV root mean square). The single-cell recordings were performed with a glass micropipette (impedance 1–10 MΩ) filled with 3 M NaCl, mounted on a micromanipulator (MO-10, Narishige, Tokyo, Japan). The electrode was slowly lowered at ~2 µm/s and spikes were detected based on visual (2002 C and 2004 C, Tektronix, OR., USA) and auditory (Audio monitor 3300, A-M Systems, WA., USA) cues. Once the spiking activity was detected, the micropipette tip was advanced slowly to properly isolate and identify single neuron activity in the recorded signal.

## Juxtacellular labeling

The procedure was similar to that of single-cell recordings except that the micropipette was filled with 2% Neurobiotin (SP-1120, Vector Laboratories, CA., USA) in 0.5 M NaCl, the tip was immersed in Dil (Vybrant Dil cell-labeling, V22885, Thermo Fisher Scientific, Mass., USA) and the impedance was periodically checked to assure that it was between 4 and 12 MΩ. With the headstage in extracellular mode and after single-cell activity was isolated, the micropipette tip was advanced until the negative spikes (extracellular recording) became positive spikes (juxtacellular recording) with an amplitude of at least 600 μV. Then, the headstage was switched to intracellular mode to juxtacellularly label the neuron following the method described by *Pinault, 1996*. The firing rate of recorded neurons were modulated by passing positive current pulses (200ms ON/OFF) at increasing intensities (1–10 nA) through the micropipette tip. After a delay of a few seconds, the electrical properties of the recorded neuron suddenly and significantly changed, increasing its firing rate and broadening the spike waveform. From this critical moment, pulse intensity was lowered to prevent cellular damage and the modulation was maintained from several seconds to minutes in order to fill the neuron with neurobiotin.

## Neuropixels recording

All Neuropixels recordings were performed using SpikeGLX (http://billkarsh.github.io/SpikeGLX/) on a computer connected to the PXIe acquisition module. Action potentials were band filtered between 0.3 and 10 kHz and sampled at 30 kHz whereas simultaneous LFPs were band filtered between 0.5 and 500 Hz and sampled at 2.5 kHz. Neuropixels's probe was coated with Dil lipophilic dye before insertion in the brain so a precise mark of the recording tract would be visible at confocal microscope. The probe was lowered in the coronal plane at 90 degrees from horizontal plane at ~2 μm/s until 4000 μm below cerebellar cortex surface. Neuropixels's probe was left to settle for 10 min to avoid drift during the recording.

## Transcranial electrical stimulation

The different protocols for transcranial currents were designed in Spike2 (Cambridge Electronic Design, CED, Cambridge, U.K.) and sent to a battery-driven linear stimulus isolator (WPI A395, Fl., USA) through an analog output from the acquisition board (CED micro1401-3). tES was applied between the ring-electrode and a reference electrode consisting of a rubber rectangular pad (6 cm$^2$) attached to the back of the mice and moisten with electrogel (Electro-Cap International, OH., USA).

To measure the actual voltage changes elicited intracranially, sinusoid alternating current waves were delivered at amplitudes of $\pm2, \pm20$ and $\pm200$ μA ($\pm0.0426, \pm0.426$ and $\pm4.26$ mA/cm$^2$) at 1 Hz and recorded in steps of 1 mm from cortical surface to 4 mm depth. We selected tACS to measure electric field strength due to two main reasons: (1) amplifiers used in electrophysiology filter out low-frequency signals like DC, making voltage changes from tDCS undetectable, and (2) DC stimulation can alter electrode and tissue impedance over time, whereas alternating the polarity in tACS helps maintain stable recordings.

To characterize the effects induced by tDCS, trials of 15 or 20 s pulses at 100, 200, and 300 μA anodal and cathodal tDCS (including 5 s ramp-in and 5 s ramp-out) were applied separated by 10 s of non-stimulation. For every single-cell recording, the maximum number of trials allowed by the recording stability were applied. A neuron was included for analysis if the recording was stable for at least three trials at a given intensity and polarity, and up to a maximum of 1 hr recording.

## Histology

To reconstruct the neurobiotin-labeled neurons, mice were deeply anesthetized with ketamine–xylazine mixture (Ketaset, 100 mg/ml; Rompun, 20 mg/ml) 15 min after juxtacellular labeling and perfused transcardially with 0.9% saline followed by 4% paraformaldehyde (Panreac, Barcelona, Spain) in PBS (0.1 M, pH ~7,4). The brains were removed and stored in 4% paraformaldehyde for 24 hours, cryoprotected in 30% sucrose in PBS the next 48 hours, and then cut into 50 μm coronal slices with a freezing microtome (CM1520, Leica, Wetzlar, Germany). After three washes with PBS-Triton X-100 1% (PBS-Tx, Sigma-Aldrich, Mo., USA), sections containing neurobiotin-labelled neurons were blocked with 10% Normal Donkey Serum (NDS, 566460, Merck, Darmstadt, Germany) in PBS-Tx and then incubated overnight at room temperature in darkness with Streptavidin 1:200 (Streptavidin DyLight 488 conjugated, Thermo Fisher Scientific) in PBS-Tx. After three washes with PBS, sections were mounted

on glass slides and coverslipped using Dako fluorescence mounting medium (Agilent Technologies, Santa Clara, CA, USA). To determine recording regions across cerebellar tissue in Neuropixels recordings, the same process was carried out with the exception that after the three washes in PBS-Triton X-100 1%, slices were incubated for 3 min with Hoechst 33342 dye (Merck Millipore, Billerica, MA, USA; 2 µg/ml) in PBS with 0.25% Triton X-100. For confocal imaging, an in vivo confocal microscope (A1R HD25, Nikon, Tokyo, Japan) was used. Z-series of optical sections (0.5 µm apart) were obtained using the sequential scanning mode.

## Data analysis

### Data collection
Spike activity was recorded with a glass micropipette or a Neuropixels probe and sampled at 25 (CED micro1401-3) or 30 kHz (IMEC-PXIe) with an amplitude resolution of 12 and 10 bits, respectively. When necessary, LFP were sampled at 2.5 kHz and the remaining non-neuronal activities (tES, juxtacellular injected currents and wheel movement) were sampled at 5 kHz.

### Intracranial electric field analysis
The peak-to-peak amplitude (electric potential) of the LFP oscillations induced by tACS, were averaged for a given intensity and depth. For every intensity, the electric field strength (differences between potentials) was calculated by computing the difference in peak-to-peak values between two consecutive depths (1 mm in distance).

### Single-cell activity
For glass micropipette recordings, only well isolated neurons, with high signal-to-noise ratio (at least four times the standard deviation of background noise) were considered for analysis. Spikes were detected offline in Spike2 (CED) and exported to Matlab 2015a (MathWorks Inc, MA., USA) software for analysis. Trials where the mouse was running were removed from analysis. For spike detection, a 'DC remove' process (time constant (s): 0.001–0.0004) was applied to reduce DC level drifts, and spikes were detected based on threshold-crossing algorithm of Spike2 software. After that, the DC remove process was carried out, and all spikes were visually confirmed and PC identified as such if CS were observed and had at least a 10–40ms pause in SS after CS occurrence. For Neuropixels recordings, channels showing PCs, with CS followed by a SS silence, were manually selected on SpikeGLX (http://billkarsh.github.io/SpikeGLX/) and exported to analyze on Spike2 (CED, Cambridge, UK). Spike sorting was carried out using the Spike2 software and spikes with similar waveforms were grouped together in the same templates. Putative PC templates were subsequently curated to exclude contamination produced by other units. For this purpose, only periods where the template amplitude was stable were used, and events (spikes) with an amplitude deviation greater than one third of the average template amplitude were excluded. Additionally, the autocorrelogram was checked to discard contaminated templates with violations of the refractory period. Clusters corresponding to putative SS and CS were identified due to their waveform and their firing frequency (<3 Hz for CS and >50 Hz for SS) and regularity, producing characteristic shoulders in the SS autocorrelogram. Finally, only those SS and CS from PCs unambiguously identified by the pause in their cross-correlogram, were used for the analysis. Subsequently, the activity of each neuron was analyzed using MATLAB. The 5 s window immediately before a stimulation ramp-in and immediately after a ramp-out were used for control and post-stimulation conditions, while the 5 s window immediately after the stimulation ramp reached the peak intensity was considered for tDCS condition. Averaged SS and CS waveforms, SS frequency, CS frequency and latency of the SS pause after CS were computed and analyzed for each condition. For SS firing rate analysis, all the trials with a given tDCS intensity and duration were averaged and then binned in 100ms epochs in the five second windows computed for statistical analysis. For CS firing rate and latency of the SS pause after a CS analysis, the procedure was the same as before but instead of averaging between trials, we computed the different parameters for the 5 s windows (before, during, and after tDCS) for every trial and the statistical comparisons were made between all the trials with a given tDCS intensity and time. Peristimulus time histograms showing the number of spikes per bin (bin size: 0.1 s for SS and 1 s for CS) were aligned with tDCS ramp-in, normalized and standardized (Z-score=X-µ/σ, where X is the firing rate at each bin, and µ and σ are the average and standard deviation, respectively, of the 5 s control window) with respect to

the average frequency of the five seconds before anodal and cathodal tDCS ramp-in. To compare the strength of the modulation for the same neuron with different tDCS intensities and between neurons, the different parameters during tDCS were normalized by their values during control condition.

### Neurobiotin-labeled neurons

Confocal images were processed in ImageJ (https://imagej.nih.gov/ij/) with the image processing package Fiji (http://fiji.sc/Fiji) to obtain a z-stack reconstruction of the neurobiotin-labelled neurons. The deviation of the somatodendritic axis with respect to the active electrode was calculated by measuring the angle between the neuronal axis and an imaginary line perpendicular to the active electrode.

### Statistical analysis

Statistical comparison was conducted using Matlab 2015a (MathWorks Inc). Normality was assessed using the Shapiro–Wilk test (p-value >0.05). Z-scored neural activity from each neuron was compared between the 5 s control, the 5 s tDCS and the 5 s post-stimulation periods by repeated measures ANOVA (RM-ANOVA) with post hoc Tukey's test for multiple comparisons. The non-parametric Friedman test was applied for comparisons when data did not permit normality assumption followed by Nemenyi post hoc test. The results are shown as mean ± SEM. Statistical significance was set at $p < 0.05$ in all cases.

## Acknowledgements

This work was supported by grants from the Spanish MINECO-FEDER (BFU2014-53820-P and BFU2017-89615-P) and FET European Union's Horizon 2020 research and innovation program (grant agreement No 101017716) to JM-R and from the US National Institutes of Health (R01MH093727 & R01NS112917) to JFM and (RF1MH114269) to JFM and JM-R. CAS-L was in receipt of an FPU grant from the Spanish Government (FPU13/04858). GS-GC was in receipt of an FPU grant from the Spanish Government (FPU21/01025).

## Additional information

### Funding

| Funder | Grant reference number | Author |
| --- | --- | --- |
| MINECO-FEDER | BFU2014-53820-P | Javier Márquez-Ruiz |
| MINECO-FEDER | BFU2017-89615-P | Javier Márquez-Ruiz |
| Horizon 2020 - Research and Innovation Framework Programme | 10.3030/101017716 | Javier Márquez-Ruiz |
| National Institutes of Health | RF1MH114269 | Javier F Medina<br>Javier Márquez-Ruiz |
| National Institutes of Health | R01MH093727 | Javier F Medina |
| National Institutes of Health | R01NS112917 | Javier F Medina |
| FPU grant from the Spanish Government | FPU13/04858 | Carlos A Sánchez-León |
| FPU grant from the Spanish Government | FPU21/01025 | Guillermo Sánchez-Garrido Campos |

The funders had no role in study design, data collection and interpretation, or the decision to submit the work for publication.

## Author contributions

Carlos A Sánchez-León, Conceptualization, Data curation, Formal analysis, Investigation, Methodology, Writing – original draft, Writing – review and editing; Guillermo Sánchez-Garrido Campos, Marta Fernández, Data curation, Formal analysis, Investigation, Writing – review and editing; Álvaro Sánchez-López, Javier F Medina, Methodology, Writing – review and editing; Javier Márquez-Ruiz, Conceptualization, Data curation, Formal analysis, Funding acquisition, Investigation, Methodology, Writing – original draft, Project administration, Writing – review and editing

## Author ORCIDs

Carlos A Sánchez-León  https://orcid.org/0000-0002-5278-9758
Guillermo Sánchez-Garrido Campos  https://orcid.org/0009-0006-8469-8160
Marta Fernández  https://orcid.org/0000-0003-4100-8693
Álvaro Sánchez-López  https://orcid.org/0000-0002-6253-1064
Javier F Medina  https://orcid.org/0000-0001-8708-5315
Javier Márquez-Ruiz  https://orcid.org/0000-0002-8036-8707

## Ethics

This study was conducted in strict accordance with the European Union guidelines (Directive 2010/63/EU) and the Spanish regulations (Royal Decree 53/2013) for the care and use of laboratory animals in research. All experimental procedures were approved by the Ethics Committee for Animal Research of Pablo de Olavide University (Seville, Spain) (Permit Numbers: 18-11-14-156; 09/05/2018/085; 06-07-2021-107).All efforts were made to minimize animal suffering, and surgical procedures were performed under anesthesia using a ketamine-xylazine mixture. Mice were monitored throughout the experiments to ensure their well-being, and proper post-surgical care was provided in compliance with institutional guidelines.

Reviewer #1 (Public review): https://doi.org/10.7554/eLife.100941.3.sa1
Reviewer #2 (Public review): https://doi.org/10.7554/eLife.100941.3.sa2
Reviewer #3 (Public review): https://doi.org/10.7554/eLife.100941.3.sa3
Author response https://doi.org/10.7554/eLife.100941.3.sa4

# Additional files

## Supplementary files

MDAR checklist

## Data availability

The data supporting the main findings of this study are publicly available at https://doi.org/10.5281/zenodo.15051403.

The following dataset was generated:

| Author(s) | Year | Dataset title | Dataset URL | Database and Identifier |
| --- | --- | --- | --- | --- |
| Sánchez-León CA, Sánchez-Garrido Campos G, Fernández M, Sánchez-López A, Medina J, Márquez-Ruiz J | 2025 | Data for "Somatodendritic orientation determines tDCS-induced neuromodulation of Purkinje cell activity in awake mice" | https://doi.org/10.5281/zenodo.15051403 | Zenodo, 10.5281/zenodo.15051403 |

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
