## [Editor Report · eLife Assessment]

In this **important** and **compelling** study, Sánchez-León et al. investigate the effects of tDCS on the firing of single cerebellar neurons in awake and anesthetized mice. They find heterogeneous responses depending on the orientation of the recorded Purkinje cell. The paper may well explain part of the controversial and ambiguous outcomes of various clinical trials.

---

## [Referee Report · Reviewer #1 (Public review)]

Summary:

In this elegant and thorough study, Sánchez-León et al. investigate the effects of tDCS on the firing of single cerebellar neurons in awake and anesthetized mice. They find heterogeneous responses depending on the orientation of the recorded Purkinje cell.

Strengths:

The paper is important in that it may well explain part of the controversial and ambiguous outcomes of various clinical trials. It is a well-written paper on a deeply analyzed dataset.

Weaknesses:

The sample size could be increased for some of the experiments.

Comments on revised version: They have not been able to increase the size of some of the critical experiments, but they have done additional statistics, which make me feel confident that the main conclusions are correct.

---

## [Referee Report · Reviewer #2 (Public review)]

Summary:

In this study by Sánchez-León and colleagues, the authors attempted to determine the influence of neuronal orientation on the efficacy of cerebellar tDCS in modulating neural activity. To do this, the authors made recordings from Purkinje cells, the primary output neurons of the cerebellar cortex, and determined the inter-dependency between the orientation of these cells and the changes in their firing rate during cerebellar tDCS application.

Strengths:

(1) A major strength is the in vivo nature of this study. Being able to simultaneously record neural activity and apply exogenous electrical current to the brain during both an anesthetized state and during wakefulness in these animals provides important insight into physiological underpinnings of tDCS.

(2) The authors provide evidence that tDCS can modulate neural activity in multiple cell types. For example, there is a similar pattern of modulation in Purkinje cells and non-Purkinje cells (excitatory and inhibitory interneurons). Together, these data provide wholistic insight into how tDCS can affect activity across different populations of cells, which is important implications for basic neuroscience, but also clinical populations where there may be non-uniform or staged effects of neurological disease on these various cell types.

(3) There is systematic investigation into the effects of tDCS on neural activity across multiple regions of the cerebellum. The authors demonstrate that the pattern of modulation is dependent on the target region. These findings have important implications for determining the expected neuromodulatory effects of tDCS when applying this technique over different target regions non-invasively in animals and humans.

(4) The authors provide a thorough background, rationale, and interpretation regarding the expected and observed influence of neuronal orientation on excitability modulation by electrical stimulation.

---

## [Referee Report · Reviewer #3 (Public review)]

Summary:

In this study, Sanchez-Leon et al. combined extracellular recordings of Purkinje cell activity in awake and anesthesized mice with juxtacellular recordings and Purkinje cell staining to link Purkinje cell orientation to their stimulation response. The authors find a relationship between neuron orientation and firing rate, dependent on stimulation type (anodal/cathodal). They also show effects of stimulation intensity and rebound effects.

Strengths:

Overall, the work is methodologically sound and the manuscript well written. The authors have taken great care to explain their rationale and methodological choices.

Weaknesses:

My only reservation is the lack of reporting of the precise test statistics, p-values and multiple comparison corrections. The work would benefit from adding this and other information.

---

## [Author Response]

The following is the authors’ response to the original reviews.

**Public Reviews:**

**Reviewer #1 (Public review):**
Summary:In this elegant and thorough study, Sánchez-León et al. investigate the effects of tDCS on the firing of single cerebellar neurons in awake and anesthetized mice. They find heterogeneous responses depending on the orientation of the recorded Purkinje cell.Strengths:The paper is important in that it may well explain part of the controversial and ambiguous outcomes of various clinical trials. It is a well-written paper on a deeply analyzed dataset.

We sincerely thank Reviewer #1 for their positive feedback and insightful comments. We are pleased to know that you found our study elegant and thorough, and we appreciate your recognition of its potential to clarify the controversial and ambiguous outcomes seen in various clinical trials. Your acknowledgment of the depth of our analysis and the clarity of the writing is highly encouraging, and we are grateful for your thoughtful evaluation of our work.

Weaknesses:The sample size could be increased for some of the experiments.

We sincerely thank the reviewer for their thoughtful suggestion to increase the sample size. While we understand the importance of this consideration, we believe it is not feasible at this stage due to several factors. First, the complexity of our experiments, which include single-neuron recordings in awake animals during electric field application, juxtacellular neurobiotin injections post-tDCS (with a low success rate), and high-density recordings from Purkinje cells across different layers in awake animals, significantly limits the throughput of data collection. Second, the statistical outcomes obtained from our analyses, which combine multiple techniques, are robust and provide a strong basis for our conclusions. Third, the current study already involves a substantial number of animals (74 mice), which aligns with ethical considerations for minimizing animal use while ensuring robust results.

We believe that the current sample size is sufficient to support the findings presented in the manuscript. Expanding the sample size further would require considerable additional resources and time, without a clear indication that it would fundamentally alter the conclusions of the study. We are grateful for the reviewer’s understanding of these limitations and their acknowledgment of the value of the current dataset.

**Reviewer #2 (Public review):**
Summary:In this study by Sánchez-León and colleagues, the authors attempted to determine the influence of neuronal orientation on the efficacy of cerebellar tDCS in modulating neural activity. To do this, the authors made recordings from Purkinje cells, the primary output neurons of the cerebellar cortex, and determined the inter-dependency between the orientation of these cells and the changes in their firing rate during cerebellar tDCS application.Strengths:(1) A major strength is the in vivo nature of this study. Being able to simultaneously record neural activity and apply exogenous electrical current to the brain during both an anesthetized state and during wakefulness in these animals provides important insight into the physiological underpinnings of tDCS.(2) The authors provide evidence that tDCS can modulate neural activity in multiple cell types.For example, there is a similar pattern of modulation in Purkinje cells and non-Purkinje cells (excitatory and inhibitory interneurons). Together, these data provide wholistic insight into how tDCS can affect activity across different populations of cells, which has important implications for basic neuroscience, but also clinical populations where there may be non-uniform or staged effects of neurological disease on these various cell types.(3) There is a systematic investigation into the effects of tDCS on neural activity across multiple regions of the cerebellum. The authors demonstrate that the pattern of modulation is dependent on the target region. These findings have important implications for determining the expected neuromodulatory effects of tDCS when applying this technique over different target regions noninvasively in animals and humans.

We sincerely thank Reviewer #2 for their detailed and thoughtful comments on our study. We are pleased that you recognized the importance of our in vivo approach, allowing for simultaneous neural recordings and tDCS application in both anesthetized and awake states. Your acknowledgment of our findings regarding the modulation of neural activity across different cell types, including Purkinje and non-Purkinje cells, is greatly appreciated. We also value your recognition of the implications of our work for understanding how tDCS can affect diverse neuronal populations, particularly in the context of clinical applications. Additionally, your positive feedback on our systematic investigation across multiple cerebellar regions highlights the relevance of our work for determining the region-specific effects of tDCS. Thank you for your encouraging and insightful evaluation.

Weaknesses:(1) In the introduction, there is a lack of context regarding why neuronal orientation might be a critical factor influencing the responsiveness to tDCS. The authors allude to in vitro studies that have shown neuronal orientation to be relevant for the effects of tDCS on neural activity but do not expand on why this might be the case. These points could be better understood by informing the reader about the uniformity/non-uniformity of the induced electric field by tDCS. In addition, there is a lack of an a priori hypothesis. For example, would the authors have expected that neuronal orientation parallel or perpendicular to the electrical field to be related to the effects of tDCS on neural activity?

We thank the Reviewer #2 for this insightful comment. In response, we have expanded the introduction to provide a clearer context regarding the influence of neuronal orientation on the effects of tDCS. Therefore, we have added two new paragraphs in the Introduction to address these points.

“For neurons whose somatodendritic axis is aligned with the electric field, the field induces a pronounced somatic polarization. In the case of anodal stimulation, where the positive electrode is positioned near the dendrites and the soma is oriented away, positively charged ions accumulate near the soma, leading to depolarization and increased excitability, thus facilitating action potential generation. Conversely, neurons whose orientation opposes the field, such as when the soma is closer to the positive electrode and the dendrites face away, experience hyperpolarization, reducing excitability. Lastly, neurons oriented perpendicular to the electric field would exhibit minimal somatic polarization, as the field does not induce significant redistribution of charges along the somatodendritic axis.”

Additionally, we have now clarified our a priori hypothesis regarding neuronal orientation and its expected influence on tDCS efficacy.

“We hypothesized that the orientation of PCs relative to the electric field would influence the effects of tDCS on neural activity. In the Vermis, PCs oriented parallel to the field are expected to exhibit stronger effects due to greater somatic polarization, leading to depolarization or hyperpolarization depending on the orientation of the somatodendritic axis. Conversely, PCs in Crus I/II, which are oriented obliquely to the field, are expected to exhibit intermediate effects, as the oblique alignment reduces the strength of polarization compared to parallel alignment.”

(2) It is unclear how specific stimulation parameters were determined. First, how were the tDCS intensities used in the present experiments determined/selected, and how does the relative strength of this induced electric field equate to the intensities used non-invasively during tDCS experiments in humans? Second, there is also a fundamental difference in the pattern of application used here (e.g., 15 s pulses separated by 10 s of no stimulation) compared to human studies (e.g., 10-20 min of constant stimulation).

We thank Reviewer #2 for their observations. We proceed to address their concerns and included the following text in the main manuscript, Discussion section:

“We used higher values than those applied in human experiments to achieve more reliable results. As seen in Supplementary Fig. 3, neurons are modulated in a similar way for 100, 200 or 300 µA but higher intensities elicited significant changes in a greater proportion of these neurons. In addition, a previous study from our lab23 using the same methodology, showed that 100, 200 and 300 µA (eliciting from 5.9 to 125.7 V/m in the current study) were ideal to obtain reliable and robust results in neuronal modulation, while keeping animal awareness of the stimulation at a minimum level. Besides, Asan et al. has recently shown that using epidural stimulation in anesthetized rats under an electric field closer to human studies (1.5–7.5 V/m) was also able to modulate the activity of cerebellar neurons.”

In addition, we add the following text to the Results section under ‘tDCS modulates Purkinje cell activity in awake mice in a heterogeneous manner’ section:

“This protocol allows us to avoid the development of plasticity effects, which are known to require at least several minutes of tDCS administration, and to test the direct electrical modulation exerted by the externally applied currents.”

(3) In their first experiment, the authors measure the electric field strength at increasing depths during increasing stimulation intensities. However, it appears that an alternating current rather than a direct current, which is usually employed in tDCS protocols, was used. There is a lack of rationale regarding why the alternating current was used for this component. Typically, this technique is more commonly used for entraining/boosting neural oscillations compared to studies using tDCS which aim to increase or decrease neural activity in general.

We appreciate Reviewer #2’s assessment of the differences between tDCS and tACS. We will clarify this distinction. We chose tACS for measuring electric field strength for two main reasons:

• Amplifier Limitations: The amplifiers commonly used in electrophysiology are designed to filter out low-frequency components, including direct current (DC) signals, using a highpass filter. This is due to the fact that the neuronal signals of interest, such as action potentials, typically occur at higher frequencies (several Hz to kHz). Consequently, any DC signal applied is filtered out from the recordings, preventing us from measuring changes in voltage effectively.

• Impedance Changes: DC stimulation can alter the impedance of electrodes and surrounding tissue over time. To mitigate this effect and maintain stable recordings, it is advantageous to frequently alternate the polarity and intensity of the stimulation.

This next text has been included in the 'Transcranial Electrical Stimulation' section of the 'Materials and Methods' part of the manuscript:

“We selected tACS to measure electric field strength due to two main reasons: (1) amplifiers used in electrophysiology filter out low-frequency signals like DC, making voltage changes from tDCS undetectable, and (2) DC stimulation can alter electrode and tissue impedance over time, whereas alternating the polarity in tACS helps maintain stable recordings.”

It is important to note that our aim with tACS is to provide an approximation of current propagation through the tissue, rather than to exactly replicate the baseline conditions encountered during continuous tDCS stimulation.

**Reviewer #3 (Public review):**
Summary:In this study, Sanchez-Leon et al. combined extracellular recordings of Purkinje cell activity in awake and anesthetized mice with juxtacellular recordings and Purkinje cell staining to link Purkinje cell orientation to their stimulation response. The authors find a relationship between neuron orientation and firing rate, dependent on stimulation type (anodal/cathodal). They also show the effects of stimulation intensity and rebound effects.Strengths:Overall, the work is methodologically sound and the manuscript is well written. The authors have taken great care to explain their rationale and methodological choices.

We sincerely thank Reviewer #3 for their positive feedback and constructive comments regarding our study. We are pleased that you found our work methodologically sound and well written. Your acknowledgment of our efforts to explain our rationale and methodological choices is greatly appreciated. We believe that the insights gained from linking Purkinje cell orientation to their stimulation response will contribute significantly to our understanding of cerebellar function and tDCS effects. Thank you for your thoughtful evaluation of our manuscript.

Weaknesses:My only reservation is the lack of reporting of the precise test statistics, p-values, and multiple comparison corrections. The work would benefit from adding this and other information.

We sincerely thank Reviewer #3 for their valuable feedback and for highlighting an important aspect of our analysis. We agree that the inclusion of precise test statistics, p-values, and details on multiple comparison corrections would strengthen the robustness of our findings. In response to your suggestion, we have now added this information to the Results section, ensuring that all statistical tests, exact p-values, and corrections for multiple comparisons are clearly reported. We believe these additions provide greater transparency and rigor to our analysis, and we appreciate your thoughtful recommendation.

Major Comments:(1) The authors should report the exact test statistics. These are missing for all comparisons and hinder the reader from understanding what exactly was tested for each of the experiments. For example, having the exact test statistics would help better understand the non-significant differences in Figure 1h where there is at least a numeric difference in CS firing rate during tDCS.

As mentioned before, we have now included the precise test statistics for all statistical comparisons throughout the manuscript. Specifically, in the case of Supplementary Figure 1h, we have added the exact values for the comparisons of CS firing rates during tDCS, even for nonsignificant differences, to ensure transparency and to clarify the observed numerical differences. We believe these additions will help readers better interpret the data and understand the statistical underpinnings of our findings.

However, given the large amount of data analyzed, particularly related to individual neuronal activity, it is not feasible to present all of the data for each individual neuron. We have aimed to provide a comprehensive statistical summary without overwhelming the reader with an excessive amount of detailed data.

(2) Did the authors apply any corrections for multiple comparisons? Generally, it would be helpful if they could clarify the statistical analysis (which values were subjected to the tests, how many tests were performed for each question, etc.).

We appreciate the reviewer’s comment regarding the need for clarification on the statistical analysis and the application of multiple comparison corrections. In response, we have updated the main text to include all the requested information. Specifically, we have added the appropriate multiple comparison tests (Tukey's or Nemenyi) where applicable to each analysis. These corrections have been applied to ensure that the results are robust and account for the number of comparisons made. We have also clarified the specific tests used for each analysis, the values subjected to these tests, and the number of comparisons performed for each question. This information is now detailed in the Methods section under 'Statistical Analysis' for transparency and to aid in the interpretation of the results.

(3) The relationship shown in Figure 2g seems to be influenced by the two outliers. Have the authors confirmed the results using a robust linear regression method?

We agree with the reviewer that the two neurons in Figure 2g could appear as outliers. To address this, we applied the ROUT method with a stringent Q = 1% to detect potential outliers, and none were found. In addition, we have confirmed the robustness of our results by performing a complementary analysis using robust linear regression methods (e.g., M-estimators), which showed consistent findings with our original analysis. For this purpose, we used the 'Huber' loss function, which combines least squares with robustness against outliers. The regression line obtained with this method (y = -0.5650x + 157.4556) differs minimally from the originally presented value, with the p-value of the slope and the intercept being p = 1.4846x10^-4^ (t_(22)_ = -4.5740) and p = 1.1382x10^-11^ (t_(22)_ = 12.8010), respectively. Author response image 1 shows both regression fits to facilitate their comparison. These additional steps ensure the reliability of the relationship observed in the figure, even when accounting for the potential influence of the two data points.

**Author response image 1. sa4fig1:** 

(4) The authors conclude that tDCS modulates vermal PCs more than Crus I/II PCs - but they don't seem to test this statistically. It would be helpful to submit the firing rate change values to an actual statistical test to conclude this directly from the data.

We agree that it would be appropriate to apply a statistical test to determine whether there is similarity in the level of modulation. To this end, we have normalized the modulation so that all data are positive. For example, a neuron that increases or decreases its activity by 50% relative to the baseline period will be considered as having a modulation of 50% in both cases. This yields a mean modulation of 9.42% for neurons recorded in Crus I/II and 62.35% for those in the Vermis. Since the two distributions do not meet the normality assumption (Shapiro-Wilk test), we used a Mann-Whitney test, which resulted in a p-value < 0.0001, thus demonstrating a significant difference in modulation between the two cerebellar regions analyzed. We added this information to the main text. Additionally, we included a new panel in Supplementary Figure 3 (Supplementary Figure 3i) to visually represent these data.

**Reviewer #1 (Recommendations for the authors):**
I have several suggestions to further improve the paper:(1) It remains unclear how many tDCS trials were done during each single-cell recording. What were the inclusion criteria? Were tens of trials done per cell or was a cell already included if the recording was stable during a few trials? Please clarify.

For every single-cell recording, the maximum number of trials allowed by the recording stability were applied. A neuron was included in the analysis if the recording was stable for at least 2 trials at a given intensity and polarity, and up to a maximum of 1 hour recording. We introduced a paragraph in the methods section explaining this.

(2) Along the same line, could the authors show cell responses to individual consecutive trials? Do the responses change over time? For example, does a cell increase the firing rate more during early trials compared to late trials? Please clarify.

We appreciate the reviewer’s suggestion to investigate whether cell responses change over consecutive trials. In our data, when tDCS effects were observed, the changes in firing rate were evident from the very first trials in some neurons. To illustrate this, we have included Author response image 2, which shows examples of individual neuron responses (2 non-PC on the left and 2 PC on the right) across consecutive trials. Red and blue histogram bars indicate anodal and cathodal tDCS periods, respectively.

**Author response image 2. sa4fig2:** 

However, a rigorous analysis of the stimulation effect over time across trials was not feasible due to the considerable variability in the number of trials applied to different recorded neurons. This variability arose from differences in the duration for which stable recordings could be maintained.

Despite this limitation, the early responses to tDCS provide valuable insights into the immediate effects of stimulation on neuronal activity.

(3) Neurons are recorded very superficially, just below a 2 mm wide craniotomy. The temperature of the brain is likely lower than a normal physiological temperature. Did the authors consider the potential effects of temperature? Please address.

We acknowledge the reviewer's concern regarding the potential effects of temperature on the recorded neurons. While it is challenging to precisely control the temperature of the tissue in the recording area, it is important to note that the temperature conditions were consistent across both the control and stimulation phases of the experiment. This consistency ensures that any potential effects of temperature are evenly distributed across conditions, thereby minimizing its impact on the observed changes in neuronal activity. Furthermore, although the recordings are conducted 2 mm below the craniotomy, this region is continuously bathed in saline, with an additional 3 mm of fluid maintained at physiological temperature, effectively preventing dehydration and cooling of the surface tissue.

(4) More general, but along the same line, is there any effect of the depth of the recorded cells on its response to stimulations for any of the data collected in this study? Figure 1 nicely shows that there is a significant electric field at depths up to 4 mm, but do more superficial cells have stronger/weaker responses to cathodal/anodal stimulation, as the electric field there is much stronger?

We were also expecting to see some correlation between depth and degree of modulation, however, a linear regression analysis showed very low R^2^ values (see Author response images 3-6), suggesting a negligible correlation between depth of recording and neuronal activity modulation. We did this analysis for Purkinje and non-Purkinje cells separately, as well as for recordings in CrusI-II or Vermis, showing similar negative results in all cases.

**Author response image 3. sa4fig3:** 

**Author response table 1. sa4table1:** 

	Anodal 200uA	Cathodal 200uA	Anodal 100uA	Cathodal 100uA	Anodal 300uA	Cathodal 300uA
Goodness of Fit						
R squared	0.02502	0.1451	0.1759	0.1393	0.07669	0.03640
Sy.x	13.44	8.150	9.884	15.24	24.73	18.60
Is slope significantly non-zero?						
F	0.4876	3.226	0.6404	0.4854	0.2492	0.1133
DFn, DFd	1,19	1,19	1,3	1,3	1,3	1,3
P value	0.4934	0.0884	0.4821	0.5361	0.6520	0.7586
Deviation from zero?	Not Significant	Not Significant	Not Significant	Not Significant	Not Significant	Not Significant
Runs test						
Points above line			2	1	2	2
Points below line			3	4	3	3
Number of runs			4	3	3	3
P value (runs test)			0.9000	> 0.9999	0.5000	0.5000
Deviation from linearity			Not Significant	Not Significant	Not Significant	Not Significant
Equation	Y=-4.355^(**)X+18.63	Y=-6.794^(**)X+22.22	Y=-6.863^(**)X+19.39	Y=-9.211^(**)X+27.75	Y=-10.71^(**)X+43.67	Y=-5.433^(**)X+26.07

**Author response image 4. sa4fig4:** 

**Author response table 2. sa4table2:** 

	Anodal 200uA	Cathodal 200uA	Anodal 100uA	Cathodal 100uA	Anodal 300uA	Cathodal 300uA
Goodness of Fit						
R squared	0.08524	0.07646	0.02507	0.2985	0.03554	0.08955
Sy.x	40.74	24.92	90.34	23.16	117.4	26.77
Is slope significantly non-zero?						
F	4.007	3.560	0.1800	2.979	0.2579	0.6885
DFn, DFd	1,43	1,43	1,7	1,7	1,7	1,7
P value	0.0517	0.0660	0.6841	0.1280	0.6271	0.4341
qquad	Not Significant	Not Significant	Not Significant	Not Significant	Not Significant	Not Significant
Runs test						
Points above line			1	3	3	3
Points below line			8	6	6	6
Number of runs			3	7	6	6
P value (runs test)			> 0.9999	> 0.9999	0.8810	0.8810
Deviation from linearity			Not Significant	Not Significant	Not Significant	Not Significant
Equation	Y=11.91^(**)X+4.489	Y=6.869^(**)X+10.01	Y=29.18^(**)X-57.44	Y=30.44^(***)X-79.64	Y=45.38^(**)X-72.17	Y=16.91^(**)X-23.60

**Author response image 5. sa4fig5:** 

**Author response table 3. sa4table3:** 

	Anodal 200uA	Cathodal 200uA	Anodal 100uA	Cathodal 100uA	Anodal 300uA	Cathodal 300uA
Goodness of Fit						
R squared	0.02341	0.02422	0.0007774	0.03864	0.005197	5.765e-006
Sy.x	43.85	38.85	93.44	34.07	45.65	65.50
Is slope significantly non-zero?						
F	0.6713	0.7199	0.02023	1.085	0.1306	0.0001499
DFn, DFd	1,28	1,29	1,26	1,27	1,25	1,26
P value	0.4195	0.4031	0.8880	0.3068	0.7208	0.9903
Deviation from zero?	Not Significant	Not Significant	Not Significant	Not Significant	Not Significant	Not Significant
Runs test						
Points above line						
Points below line						
Number of runs						
P value (runs test)						
qquad						
Equation	Y=-7.697^(**)X+81.16	Y=-6.829^(**)X+62.28	Y=-2.886^(**)X+58.77	Y=-7.682^(**)X+47.56	Y=3.645^(**)X+78.34	Y=0.1767^(**)X+76.78

**Author response image 6. sa4fig6:** 

**Author response table 4. sa4table4:** 

	Anodal 200uA	Cathodal 200uA	Anodal 100uA	Cathodal 100uA	Anodal 300uA	Cathodal 300uA
Goodness of Fit						
R squared	0.02210	0.08475	0.04235	7.561e-005	0.05198	0.05177
Sy. X	129.9	61.06	28.04	26.90	79.95	87.77
Is slope significantly non-zero?						
F	0.5199	2.130	0.8845	0.001588	1.097	1.092
DFn, DFd	1,23	1,23	1,20	1,21	1,20	1,20
P value	0.4782	0.1580	0.3582	0.9686	0.3075	0.3085
qquad	NotSignificant	Not Significant	NotSignificant	Not Significant	Not Significant	Not Significant
Runs test						
Points above line						
Points below line						
Number of runs						
P value (runs test)						
qquad						
Equation	Y=-20.88^(**)X+113.8	Y=19.87^(**)X+24.76	Y=-5.902^(**)X+44.05	Y=0.2396^(**)X+32.62	Y=19.52^(**)X+54.69	Y=20.83^(**)X+33.52

(5) The authors are recording the movements of the mouse on a treadmill. Was there any correlation between tDCS and behavior? And between behavior and firing patterns? Please address.

We appreciate the reviewer’s question regarding the potential correlation between tDCS and behavior, as well as between behavior and firing patterns. In our experimental setup, the movement of the mouse typically introduces electrical artifacts in the recordings, particularly during running on the treadmill. To ensure the accuracy of our data, trials that coincided with running or other significant movements were excluded from the analysis. This is explained in the Methods section of the main text under 'Data analysis' within the description of how single-cell activity was processed. On the other hand, conscious of the modulatory effects that animal movement or specific behaviors may have on neuronal firing rates, we thought that trials involving movement should be eliminated to avoid any potential confounding with the effects of current application.

(6) The strength of the electrical field seems highly variable. Do the authors have an explanation for this? Please address.

We appreciate the reviewer’s observation regarding the variability in the strength of the electric field. This variability is indeed expected, given the inherent inter-individual differences in skull thickness across animals (which, as discussed in the main manuscript, attenuates around 20% of the current), as well as slight variations in the precise placement of the tES active electrode during surgery. These factors can lead to fluctuations in the electric field, although they remain within the same order of magnitude.

(7) As the authors stated, even for cells recorded at a depth of over 2 mm, the electric fields are still much higher than the fields generated in human studies. Why were there no comparable strengths used? Please address.

We thank the reviewer for raising this important point. Previous studies from our lab (SánchezLeón et al. 2021) demonstrated minimal modulation in neuronal activity (LFP) when using tDCS intensities below 200 µA in awake animals. To achieve stronger and more consistent effects, we selected an intensity of 200 µA for our experiments. It is well-established that small animals, such as mice, require higher electric field strengths than humans to induce observable effects (Ozen et al., 2010; Vöröslakos et al., 2018; Asan et al., 2020). This discrepancy may be attributed to several factors, including differences in neuronal density within the stimulated networks (Herculano-Houzel et al., 2009), as well as variations in axonal length and diameter (Chakraborty et al., 2018). However, as we stated in the Discussion, we also found modulated neurons for electric fields close to those in humans:

“Importantly, we observe clear firing rate modulation of PCs and non-PCs at depths of 2.3 mm and tDCS intensity of 100 μA, where the measured electric field is as low as 5.9 V/m.”

Despite these limitations, animal models remain invaluable for obtaining high-resolution invasive data that cannot be collected in human studies. Such experiments are crucial for understanding the basic mechanisms underlying non-invasive brain stimulation, validating computational models, and exploring the therapeutic potential of these techniques for various neurological conditions.

References:

Asan, A. S., Lang, E. J., & Sahin, M. (2020). Entrainment of cerebellar purkinje cells with directional AC electric fields in anesthetized rats. Brain stimulation, 13(6), 1548–1558. https://doi.org/10.1016/j.brs.2020.08.017

Chakraborty, D., Truong, D. Q., Bikson, M., & Kaphzan, H. (2018). Neuromodulation of Axon Terminals. Cerebral cortex (New York, N.Y. : 1991), 28(8), 2786–2794. https://doi.org/10.1093/cercor/bhx158

Herculano-Houzel S. (2009). The human brain in numbers: a linearly scaled-up primate brain. Frontiers in human neuroscience, 3, 31. https://doi.org/10.3389/neuro.09.031.2009

Ozen, S., Sirota, A., Belluscio, M. A., Anastassiou, C. A., Stark, E., Koch, C., & Buzsáki, G. (2010). Transcranial electric stimulation entrains cortical neuronal populations in rats. The Journal of neuroscience : the official journal of the Society for Neuroscience, 30(34), 11476–11485. https://doi.org/10.1523/JNEUROSCI.5252-09.2010

Vöröslakos, M., Takeuchi, Y., Brinyiczki, K., Zombori, T., Oliva, A., Fernández-Ruiz, A., Kozák, G., Kincses, Z. T., Iványi, B., Buzsáki, G., & Berényi, A. (2018). Direct effects of transcranial electric stimulation on brain circuits in rats and humans. Nature communications, 9(1), 483. https://doi.org/10.1038/s41467-018-02928-3

(8) It seems that there is a very high number of mice used for a relatively small number of cellular recordings. Can the authors explain this?

We appreciate the reviewer’s observation regarding the number of mice used relative to the number of recorded neurons. There are several factors contributing to this:

(1) In vivo juxtacellular labeling is a complex, multi-step process where each step must be executed precisely to successfully label a neuron. During blind recordings, it is impossible to ensure with 100% certainty that the neuron targeted for juxtacellular labeling will later be recoverable with sufficient staining (Pinault, 1996). To maintain confidence in the correspondence between the recorded and labeled neuron, we typically limit our attempts to label one neuron per mouse, or at most, two neurons located far apart from each other.

(2) Recording duration limitations: The probability of maintaining a well-isolated, stable neuronal recording decreases significantly as the recording time increases. To obtain sufficient data with multiple tDCS trials, it is necessary to conduct numerous independent recordings. Additionally, each time the recording pipette penetrates the recording site, there is a minor but cumulative impact on the dura mater and neural tissue, leading to tissue degradation in subsequent recordings.

(3) Diverse experimental conditions: This study explores several conditions, including recordings in anesthetized and awake mice, targeting different cerebellar regions (Crus I/II and vermis), and utilizing a range of techniques (single-unit extracellular recordings using glass pipettes, juxtacellular recording and labeling, and high-density recordings using the Neuropixels system). These distinct approaches required the establishment of independent experimental animal groups, which contributed to the higher number of subjects used in the study.

Although we were often able to record several neurons per mouse, the final number of neurons that met all criteria for analysis was reduced due to these limitations.

References:

Pinault D. (1996). A novel single-cell staining procedure performed in vivo under electrophysiological control: morpho-functional features of juxtacellularly labeled thalamic cells and other central neurons with biocytin or Neurobiotin. Journal of neuroscience methods, 65(2), 113–136. https://doi.org/10.1016/0165-0270(95)00144-1

(9) The N for both the neurobiotin-stained neurons and the Neuropixels recordings was relatively low. If possible, it would be nice to see a few more cells.

We sincerely thank the reviewer for their thoughtful suggestion to increase the sample size. While we understand the importance of this consideration, we believe it is not feasible at this stage due to several factors. First, the complexity of our experiments, which include single-neuron recordings in awake animals during electric field application, juxtacellular neurobiotin injections post-tDCS (with a low success rate), and high-density recordings from Purkinje cells across different layers in awake animals, significantly limits the throughput of data collection. Second, the statistical outcomes obtained from our analyses, which combine multiple techniques, are robust and provide a strong basis for our conclusions. Third, the current study already involves a substantial number of animals (74 mice), which aligns with ethical considerations for minimizing animal use while ensuring robust results.

We believe that the current sample size is sufficient to support the findings presented in the manuscript. Expanding the sample size further would require considerable additional resources and time, without a clear indication that it would fundamentally alter the conclusions of the study. We are grateful for the reviewer’s understanding of these limitations and their acknowledgment of the value of the current dataset.

(10) tDCS and tES seem to be used interchangeably; please make it consistent.

We agree that this could cause confusion. To address this, we have added a clarification at the first mention of tES in the manuscript, indicating that tES (transcranial Electrical Stimulation) is an umbrella term that encompasses both tDCS (transcranial Direct Current Stimulation) and tACS (transcranial Alternating Current Stimulation). We have ensured consistent use of the appropriate term throughout the rest of the text.

(11) Did the authors apply saline or agar to the craniotomy while recording? Or was the dura dried out? Can the authors clarify this, and relate the answer to a potential interaction of either the medium or dryness of the dura with the tDCS?

We appreciate the reviewer’s inquiry. To prevent the dura from drying out during our recordings, we applied saline to the cranial window throughout the experiment. Additionally, in our setup, the tDCS ring-shaped electrode was placed over the skull and sealed with dental cement to prevent any leakage of currents into the craniotomy, which was positioned at the center of the preparation. This precaution also helped minimize electrical noise reaching the recording electrode. In instances where the seal was not perfectly executed, the electrical noise from tDCS leaked into the saline solution, causing amplifier saturation and rendering neuronal activity recordings impossible.

(12) There are several mistakes in spelling and grammar throughout the document; please check carefully.

We appreciate the reviewer’s attention to detail regarding spelling and grammar. We have carefully reviewed the manuscript and corrected all identified errors to ensure clarity and proper language use throughout the document.

(13) Can the authors briefly explain why tACS (and not tDCS) is used to measure the effectiveness of the stimulation at the different depths as shown in Figure 1? As the rest of the paper focuses entirely on tDCS, it is important to understand why tACS is used in Figure 1.

We will clarify this distinction. We chose tACS for measuring electric field strength for two main reasons:

• Amplifier Limitations: The amplifiers commonly used in electrophysiology are designed to filter out low-frequency components, including direct current (DC) signals, using a highpass filter. This is due to the fact that the neuronal signals of interest, such as action potentials, typically occur at higher frequencies (several Hz to kHz). Consequently, any DC signal applied is filtered out from the recordings, preventing us from measuring changes in voltage effectively.

• Impedance Changes: DC stimulation can alter the impedance of electrodes and surrounding tissue over time. To mitigate this effect and maintain stable recordings, it is advantageous to frequently alternate the polarity and intensity of the stimulation.

This next text has been included in the 'Transcranial Electrical Stimulation' section of the 'Materials and Methods' part of the manuscript:

“We selected tACS to measure electric field strength due to two main reasons: (1) amplifiers used in electrophysiology filter out low-frequency signals like DC, making voltage changes from tDCS undetectable, and (2) DC stimulation can alter electrode and tissue impedance over time, whereas alternating the polarity in tACS helps maintain stable recordings.”

It is important to note that our aim with tACS is to provide an approximation of current propagation through the tissue, rather than to exactly replicate the baseline conditions encountered during continuous tDCS stimulation.

(14) How do Figures 2e and f relate to each other? Figure 2e has 6 red lines, but 6f has 8 red explicitly states that 8 cells were recorded.

We appreciate the Reviewer for highlighting this discrepancy. You are correct that in Figure 5e, the lines are too densely packed to easily distinguish all of them. Additionally, the activity of two neurons under anodal tDCS was greatly suppressed, which caused their corresponding arrowheads to be close to the origin of the arrows, making them less visible. To clarify, while Figure 5f shows all 8 cells recorded, the compression of the data in Figure 5e makes it challenging to distinguish all individual responses visually. We have added a clarifying note to the figure legend to explaining that “densely packed lines and suppressed activity of two neurons under anodal tDCS reduce the visibility of their responses”.

(15) Figure 2g contains two outliers that seem critical to the correlation, this is noticeable as nearly all other cells seem to modulate much more modestly. Maybe add a few more cells to convince everyone?

We agree with the reviewer that the two neurons in Figure 2g could appear as outliers. To address this, we applied the ROUT method with a stringent Q = 1% to detect potential outliers, and none were found. In addition, we have confirmed the robustness of our results by performing a complementary analysis using robust linear regression methods (e.g., M-estimators), which showed consistent findings with our original analysis. For this purpose, we used the 'Huber' loss function, which combines least squares with robustness against outliers. The regression line obtained with this method (y = -0.5650x + 157.4556) differs minimally from the originally presented value, with the p-value of the slope and the intercept being p = 1.4846x10^-4^ (t_(22)_ = -4.5740) and p = 1.1382x10^-11^ (t_(22)_ = 12.8010), respectively. Author response image 1 both regression fits to facilitate their comparison. These additional steps ensure the reliability of the relationship observed in the figure, even when accounting for the potential influence of the two data points.

(16) 'From these experiments we can conclude that (1) tDCS in vermis of anesthetized mice modulates PCs and non-PCs in a heterogeneous way'. Figure 4d shows no correlation between cathodal versus anodal stimulation for non-PCs, so how does the data suggest heterogeneous modulation of non-PCs? Is it simply heterogeneous because the data is very scattered?

Thank you for your observation. By 'heterogeneous modulation,' we indeed refer to the scattered nature of the responses in non-PCs. Although Figure 4d shows a wide spread of data points and the linear regression is not statistically significant, a general trend can still be observed, where 11 out of 15 non-PCs show modulation in opposite directions with anodal and cathodal tDCS. However, this trend is not consistent across all neurons, hence our description of this modulation as heterogeneous. Importantly, this contrasts with the response observed in Purkinje cells (PCs), where a more consistent modulation pattern is evident, and the p-value for the linear regression is significant. Therefore, we conclude that while PCs show a clearer, more predictable modulation, the scattered data in non-PCs supports a more heterogeneous response.

(17) The authors state that it is not possible to discriminate the non-PCs, even though some published papers suggest this is quite possible (see e.g., work by Simpson and Ruigrok; please discuss). For sure, the authors of the current manuscript should be able to discriminate the interneurons in the molecular layer from those in the granular layer (if it were only by identifying the polarity of the complex spikes). The authors may want to consider redoing the analyses of the non-PCs, and at least present and compare the outcomes of these two main subgroups of non-PCs.

The authors are indeed familiar with the work of Simpson, Ruigrok, and others in linking electrophysiological recordings with neuronal class identity. Prior to proceeding with juxtacellular labeling, we conducted preliminary attempts to categorize non-PC neurons based on firing characteristics. However, we ultimately chose not to include neuronal sorting for non-PCs in this study for two main reasons.

First, the baseline recording period without tDCS was very short (10 seconds), and once tDCS was applied, the firing rate, coefficient of variation, and interspike intervals (ISI) of neurons were already altered. This made it difficult to reliably classify neurons based on their spontaneous activity, which is critical for precise sorting.

Second, unlike PCs—where the presence of complex spikes and the resulting inhibition provide a clear ground truth—there is no analogous, unequivocal marker for non-PCs. Even following the reviewer's suggestion, while it might be possible in the molecular layer to identify a neuron as a molecular layer interneuron (MLI), this approach does not allow for a rigorous distinction between basket cells and stellate cells. These two cell types, despite their distinct morphologies—which could significantly affect their responses to tDCS—cannot be reliably differentiated without a true ground truth. Therefore, in the absence of such definitive markers, we believe that further subclassification of non-PCs based solely on electrophysiological properties would not be sufficiently rigorous for the purposes of our study.

(18) Can the authors briefly discuss possible reasons why non-PCs in Crus1/2 do show heterogeneous responses similar to that of PCs, whereas the non-PCs in the vermis do not?

We appreciate the reviewer’s insightful question regarding the different modulation patterns observed in non-PCs between Crus I/II and the vermis. Several potential factors could contribute to these differences, including variations in local cerebellar circuit connectivity between the two regions, differences in the cellular diversity of non-PCs due to the lack of a "ground truth" for their classification, or disparities in somatodendritic orientation and cell distribution. In the vermis, PCs are organized into different layers with opposing orientations (as shown in Figure 6), which could result in a more stable, polarity-dependent modulation, making their response more distinct from that of non-PCs. In contrast, in Crus I/II, the orientation of PCs is more heterogeneous and less aligned with the electric field, potentially leading to a more variable modulation pattern in both PCs and non-PCs.

However, it is important to note that we did not aim to juxtacellularly label non-PCs in this study, so we cannot offer a definitive answer regarding their precise orientation or identity. Additionally, the observed differences could be partially attributed to statistical power: we recorded 50 nonPCs in Crus I/II compared to only 25 in the vermis. Out of the 15 neurons in the vermis that showed statistically significant modulation, 11 displayed polarity-dependent modulation in opposite directions, but the smaller sample size might have limited our ability to detect the full range of possible effects. Furthermore, recordings in Crus I/II were conducted in awake animals, whereas the neurons recorded in Figure 4 in the vermis were obtained from anesthetized animals. This difference in physiological state could also be related to the observed changes.

(19) 'The importance of PC axodendritic orientation in determining the effect of tDCS on firing rate modulation is further highlighted by our observation that pre-synaptic non-PC neurons providing inputs to PCs modulate their activity in a very heterogeneous way.' This is based on the finding that non-PCs modulate heterogeneously, but that is not what is shown for the vermis. Please address.

Thank you for pointing this out. By 'heterogeneous modulation,' we are referring to the observation that non-Purkinje cells (non-PCs) respond in various ways under tDCS. Specifically, some nonPCs increase their activity under anodal stimulation and decrease it under cathodal stimulation (and vice versa), while others exhibit more complex patterns, such as increasing their activity under both anodal and cathodal stimulation or decreasing for both polarities. Additionally, some non-PCs only respond to one polarity, and others show no response at all.

Our reasoning is that if the presynaptic non-PCs providing inputs to Purkinje cells (PCs) were the primary drivers of PC modulation, we would expect them to behave in a manner opposite to how PCs are modulated. For instance, if most non-PCs increased their activity under anodal stimulation while PCs decreased theirs, this could suggest that tDCS modulates non-PCs to fire more, imposing greater inhibition on PCs since many non-PCs are inhibitory. However, what we observe is a highly heterogeneous response from non-PCs, with no clear pattern that would consistently explain the modulation of PCs through presynaptic inputs alone. While non-PCs must certainly exert some influence on PC activity, their variable responses suggest that the modulation of PCs may also be driven by direct effects of tDCS on the PCs themselves, in addition to any indirect presynaptic influence.

(20) To help in reinforcing the hypothesis that stimulation response depends on dendritic orientation, the authors could show, with the existing data, how PCs in different layers of the vermis respond to cathodal or anodal stimulations. The data shown in Figure 4a-c already has a large number of PCs recorded in different layers of the vermis. As shown in Figure 4b, PCs in specific layers of the vermis have specific dendritic orientations. Can the authors show that PCs recorded for Figure 4, in the different layers (implying similar dendritic orientation) have similar (or different) stimulation responses? This would greatly improve their argument for the importance of dendritic orientation for tDCS responses.

We appreciate the reviewer’s suggestion and the valuable insight it provides. In fact, this was one of the main motivations for performing the experiments shown in Figure 6, where we conducted simultaneous recordings of different Purkinje cells (PCs) in distinct layers. This allowed us to directly compare responses in neurons with different somatodendritic orientations. Unfortunately, the data presented in Figure 4 were obtained using glass micropipettes for juxtacellular labeling— a method that permits recording from only one neuron at a time—thus precluding a robust analysis of the correlation between dendritic orientation and tDCS responses. Furthermore, it should be noted that Figure 4a represents an idealized approximation; since these recordings were performed in different animals with variations along the anteroposterior axis, precise dendritic orientation cannot be reliably attributed to each cell (except for those that were juxtacellularly labeled).

Additionally, unlike recordings with Neuropixels, where we have numerous contacts positioned at known distances from each other, enabling us to precisely locate cells within the cerebellar layers, the localization of neurons recorded with glass pipettes is less accurate. This is due to factors such as tissue displacement during insertion and animal movements, which further complicates the precise determination of neuronal layer placement during the stimulation protocol.

While the data in Figure 4 do not allow us to definitively test our hypothesis, the results shown in Figure 6 provide a more direct comparison of the responses of PCs across different layers to tDCS, thereby reinforcing the hypothesis that dendritic orientation is a key factor in modulating neuronal activity.

(21) The data shown in Figure 5e-f feels underpowered, although the statistical correlation between dendritic orientation and response is strong. For example, currently, the authors show that at an angle of ~0 degrees, two cells increase their firing to anodal stimulation, and 1 cell at 180 ~degrees decreases its firing. Again, the manuscript would be much improved if the authors could increase the sample sizes for these experiments.

We appreciate the reviewer’s concern regarding the sample size in Figure 5e-f. While the statistical correlation between dendritic orientation and response to tDCS is strong, we understand that the data may feel underpowered, particularly given the limited number of cells observed at specific angles such as ~0 degrees and ~180 degrees.

It’s important to note that although visually it may appear there is only one neuron at 180 degrees during anodal stimulation, there are actually three neurons at this orientation. This is more clearly visible in the same figure during cathodal stimulation. However, the firing rate of these neurons during anodal stimulation is so low that the arrow representing their response appears very small, making it difficult to distinguish. (We have added a clarifying note to the figure legend to explaining that “densely packed lines and suppressed activity of two neurons under anodal tDCS reduce the visibility of their responses”).

Unfortunately, increasing the sample size for these specific experiments is not feasible within the current study due to the technical complexity and time-consuming nature of the recordings, especially when incorporating juxtacellular labeling or high-density electrode arrays. Despite these challenges, we believe the current sample provides valuable insights into the relationship between dendritic orientation and firing rate modulation under tDCS. The significant statistical correlation suggests that the observed trend is robust, even with the existing sample size. Additionally, the different experimental approaches used in this study—single-unit extracellular recordings in different regions of the cerebellum in both awake and anesthetized animals, juxtacellular recordings and labeling, and high-density multi-unit recordings—provide a robust and comprehensive view of the results. Each technique offers complementary insights, strengthening our conclusions and ensuring that the observed patterns are not the result of one specific method or condition. Future studies could aim to expand on these findings, but we are confident that the results presented here contribute meaningfully to our understanding of how dendritic orientation influences neuronal responses to tDCS.

(22) The authors, rightly so, address the potential impact of plasticity in the discussion. Here, the authors may want to cite other studies that have directly addressed this question: E.g., Das et al., 2017 (Frontiers Neuroscience, 11:444; doi: 10.3389/fnins.2017.00444) and van der Vliet et al., 2018 (Brain Stimul, 11(4):759-771; doi: 10.1016/j.brs.2018.04.009).

We appreciate the reviewer’s suggestion to include additional studies addressing the impact of plasticity on the effects of cerebellar tDCS. In response, we have added a new sentence in the discussion section that cites both Das et al. (2017) and van der Vliet et al. (2018), highlighting the importance of synaptic plasticity in the effects of tDCS.

“These findings are consistent with previous work suggesting that synaptic plasticity is crucial for the effects of tDCS, as demonstrated by the importance of PC plasticity in behavioral outcomes(51) and the role of BDNF-mediated plasticity in motor learning(52).”

**Reviewer #2 (Recommendations for the authors):**
In the introduction, it would be beneficial to provide additional context regarding the influence of neuronal orientation on modulation shown from in-vitro studies. In addition, some explanation of the uniformity/non-uniformity of the electrical field would help. From here, the authors should provide their specific hypotheses for these experiments.

We thank the Reviewer #2 for this insightful comment. In response, we have expanded the introduction to provide a clearer context regarding the influence of neuronal orientation on the effects of tDCS. Therefore, we have added two new paragraphs in the Introduction to address these points.

“For neurons whose somatodendritic axis is aligned with the electric field, the field induces a pronounced somatic polarization. In the case of anodal stimulation, where the positive electrode is positioned near the dendrites and the soma is oriented away, positively charged ions accumulate near the soma, leading to depolarization and increased excitability, thus facilitating action potential generation. Conversely, neurons whose orientation opposes the field, such as when the soma is closer to the positive electrode and the dendrites face away, experience hyperpolarization, reducing excitability. Lastly, neurons oriented perpendicular to the electric field would exhibit minimal somatic polarization, as the field does not induce significant redistribution of charges along the somatodendritic axis.”

Additionally, we have now clarified our a priori hypothesis regarding neuronal orientation and its expected influence on tDCS efficacy.

“We hypothesized that the orientation of PCs relative to the electric field would influence the effects of tDCS on neural activity. In the Vermis, PCs oriented parallel to the field are expected to exhibit stronger effects due to greater somatic polarization, leading to depolarization or hyperpolarization depending on the orientation of the somatodendritic axis. Conversely, PCs in Crus I/II, which are oriented obliquely to the field, are expected to exhibit intermediate effects, as the oblique alignment reduces the strength of polarization compared to parallel alignment.”

Justification of the stimulation parameters used (i.e., intensity and pattern) should be included in the Methods.

The time of stimulation was chosen of only a few seconds to avoid confounding effects of plasticity, which is known to require several minutes of tDCS administration. Regarding the intensities, we refer to previous studies from our lab, using the exact same methodology, where we find that 100, 200 and 300 µA were ideal to obtain reliable and robust results in neuronal modulation, while keeping animal awareness of the stimulation at a minimum level. We also added the clarification to the main text.

Please also justify the use of tACS rather than tDCS in the first experiment.

We appreciate Reviewer #2’s assessment of the differences between tDCS and tACS. We will clarify this distinction. We chose tACS for measuring electric field strength for two main reasons:

• Amplifier Limitations: The amplifiers commonly used in electrophysiology are designed to filter out low-frequency components, including direct current (DC) signals, using a highpass filter. This is due to the fact that the neuronal signals of interest, such as action potentials, typically occur at higher frequencies (several Hz to kHz). Consequently, any DC signal applied is filtered out from the recordings, preventing us from measuring changes in voltage effectively.

• Impedance Changes: DC stimulation can alter the impedance of electrodes and surrounding tissue over time. To mitigate this effect and maintain stable recordings, it is advantageous to frequently alternate the polarity and intensity of the stimulation.

This next text has been included in the 'Transcranial Electrical Stimulation' section of the 'Materials and Methods' part of the manuscript:

“We selected tACS to measure electric field strength due to two main reasons: (1) amplifiers used in electrophysiology filter out low-frequency signals like DC, making voltage changes from tDCS undetectable, and (2) DC stimulation can alter electrode and tissue impedance over time, whereas alternating the polarity in tACS helps maintain stable recordings.”

It is important to note that our aim with tACS is to provide an approximation of current propagation through the tissue, rather than to exactly replicate the baseline conditions encountered during continuous tDCS stimulation.

**Reviewer #3 (Recommendations for the authors):**
(1) A suggestion would be to highlight which of the data points in Figure 2g are the neurons they show as representative in Figure 2e-f. This would give the reader insights into how a standard neuron would behave/how representative these neurons are.

We appreciate the reviewer’s comment and, in response, we have highlighted the two exemplary neurons from Figures 2e-f in Figure 2g to provide better insight into how these representative neurons behave in the context of the overall data. This will help the reader understand how typical these neurons are in relation to the broader dataset. Additionally, we have applied the same approach to Figure 3, highlighting the representative neurons for further clarity.

(2) It would also be interesting to add figures to the supplementary materials that show the waveforms of non-PC neurons during anodal and cathodal tDCS, as done for PC neurons in the supplementary materials (as stated at the bottom of page 14, the authors chose to mention but not show these).

We understand the reviewer’s interest in visualizing the waveforms of non-Purkinje neurons during anodal and cathodal tDCS. To address this, we have carefully examined the waveforms of both non-Purkinje neurons under these conditions. However, given the absence of notable changes in their waveforms, we believe that this data does not have sufficient standalone significance to justify the inclusion of a new figure. We are, of course, happy to provide this data upon request or to incorporate it into the supplementary materials if deemed necessary.

**Author response image 7. sa4fig7:** Superimposed averaged SS waveforms under control (black), anodal (red) and cathodal (blue) tDCS from the example neurons shown in panels A and B in Figure 3.

(3) In Figure 5d, there is a significant aftereffect of the stimulation on the Purkinje cell firing rate - do the authors have an idea why this occurred?

We appreciate the reviewer’s observation, as it highlights an interesting phenomenon that we have not been able to fully explain. We observed this aftereffect in many of the recorded neurons, and intriguingly, it often occurred in the opposite direction to the modulation seen during tDCS. We addressed a potential explanation for this in the discussion section:

‘Nonetheless, we cannot rule out the possibility of indirect synaptic effects. Indeed, the electric field gradient imposed by tDCS could indirectly modulate a specific neuron firing rate by increasing (or decreasing) its pre-synaptic activity, i.e. by modulating the firing rate of other neurons that synapse onto it. Indeed, these synaptic changes could explain the rebound effect observed after tDCS termination. The synapses involved in the modulation of firing rate may undergo a short-term plasticity process(47–50), which can continue to affect the firing rate even after the external currents have been turned off and no polarization is exerted on the neuron. These findings are consistent with previous work suggesting that synaptic plasticity is crucial for the effects of tDCS, as demonstrated by the importance of PC plasticity in behavioral outcomes(51) and the role of BDNF-mediated plasticity in motor learning(52).’

This explanation highlights the potential role of synaptic plasticity and the indirect modulation of neuronal networks, but further investigation would be required to fully understand the mechanisms underlying this aftereffect.

(4) I'm having trouble understanding the reference electrode positioning from schematics 1a & 1b: The text and 1a suggest that the reference electrode was positioned on the back of the mouse, outside of the brain. But Figure 1b looks as if the reference electrode was on the mouse cerebral cortex. Could the authors adapt schematic 1b to clarify the reference location or add this information to the legend?

We agree that the figure showing two different reference electrodes was confusing, and we have now modified it to better clarify the distinction between the recording reference electrode and the stimulation reference electrode. Additionally, we have specified in Figures 1A and 1B whether the reference pertains to the transcranial alternating stimulation or to the electrophysiological recording.

(9) In the discussion, (page 22) the authors highlight the importance of axodendritic orientation, but they analyze only somatodendritic orientation. Are the two so similar that they can be used synonymously? This would be good to clarify.

We appreciate the reviewer’s clarification and fully agree. While Purkinje cells (PCs) do indeed have a highly polarized morphology, with the axon generally oriented in the opposite direction to the main dendrites, this is not always the case, especially for other types of neurons. Therefore, our results strictly refer to the somatodendritic axis, as this is the one we can most clearly observe through our juxtacellular labeling. In response, we have changed all instances where the term 'axodendritic' appeared in the text to 'somatodendritic' for accuracy.

(10) It would be helpful to clarify that Supplementary Figure 3b and 3e are the same as Figures 4 c and 4d, respectively. This was confusing to me.

We appreciate the reviewer’s feedback and have now modified the caption of Supplementary Figure 3 to indicate that Supplementary Figures 3b and 3e correspond to Figures 4c and 4d, respectively. This should help clarify any confusion.

(11) Typo: 'consisting in' ◊ consisting of

We thank the reviewer for their clarification. The typo has been corrected to 'consisting of'.